# NovelQA: Benchmarking Question Answering on Documents Exceeding 200K Tokens

**Cunxiang Wang**[1*], **Ruoxi Ning**[12*], **Boqi Pan**[3], **Tonghui Wu**[3], **Qipeng Guo**[4], **Cheng Deng**[5],
**Guangsheng Bao**[1], **Xiangkun Hu**[5], **Zheng Zhang**[6], **Qian Wang**[3†] and **Yue Zhang**[1†]

[1]Westlake University; [2]University of Waterloo; [3]Hangzhou Normal University;
[4]Shanghai AI Lab; [5]SJTU; [6]NYU Shanghai
{wangcunxiang, zhangyue}@westlake.edu.cn; ruoxining@outlook.com; 20200142@hznu.edu.cn

## Abstract

Recent advancements in Large Language Models (LLMs) have pushed the boundaries of natural language processing, especially in long-context understanding. However, the evaluation of these models' long-context abilities remains a challenge due to the limitations of current benchmarks. To address this gap, we introduce NovelQA, a benchmark tailored for evaluating LLMs with complex, extended narratives. Constructed from English novels, NovelQA offers a unique blend of complexity, length, and narrative coherence, making it an ideal tool for assessing deep textual understanding in LLMs. This paper details the design and construction of NovelQA, focusing on its comprehensive manual annotation process and the variety of question types aimed at evaluating nuanced comprehension. Our evaluation of long-context LLMs on NovelQA reveals significant insights into their strengths and weaknesses. Notably, the models struggle with multi-hop reasoning, detail-oriented questions, and handling extremely long inputs, with average lengths exceeding 200,000 tokens. Results highlight the need for substantial advancements in LLMs to enhance their long-context comprehension and contribute effectively to computational literary analysis.

## 1 Introduction

Recent years have seen a remarkable surge in the development of Large Language Models (LLMs) (OpenAI, 2023b; Touvron et al., 2023). Among these developments, long-context LLMs stand out for their ability to process and interpret extended pieces of text (Tworkowski et al., 2023; Team, 2023; Anthropic, 2023). This capability is essential for complex tasks that require a deep and nuanced understanding of lengthy documents, such as legal cases (Xiao et al., 2021) or academic papers (Groeneveld et al., 2024), where the key is to understand extended narratives (Xu et al., 2023). In addition, the ability to analyze extremely long documents and multiple documents simultaneously is increasingly crucial, supporting more informed decision-making in various fields (Deng et al., 2023; Lin et al., 2023; Boiko et al., 2023).

The evaluation of extremely long-context capabilities presents challenges, as existing benchmarks (Yang et al., 2018; Tay et al., 2021) no longer align with the advanced processing abilities of current LLMs (Anthropic, 2023; Team, 2023; OpenAI, 2023a). The increasing context window size of LLMs now outpaces the average token lengths found in long-range datasets. This gap is evident, as the most advanced long-context LLMs are capable of processing over 250,000 tokens, a stark contrast to the longest average token length in current benchmarks, which is around 60,000 tokens. This mismatch underscores the need for updated evaluation methods that can accurately reflect the capabilities of current and future LLMs, as illustrated in Figure 1.

To fill this gap, we introduce NovelQA, a benchmark crafted to specifically evaluate LLMs' performance on texts with averaged context windows exceeding 200,000 tokens. Unlike existing

---

[*] Equal contribution, paper finished at Westlake University
[†] Correspondence to Yue Zhang (zhangyue@westlake.edu.cn) and Qian Wang (20200142@hznu.edu.cn).

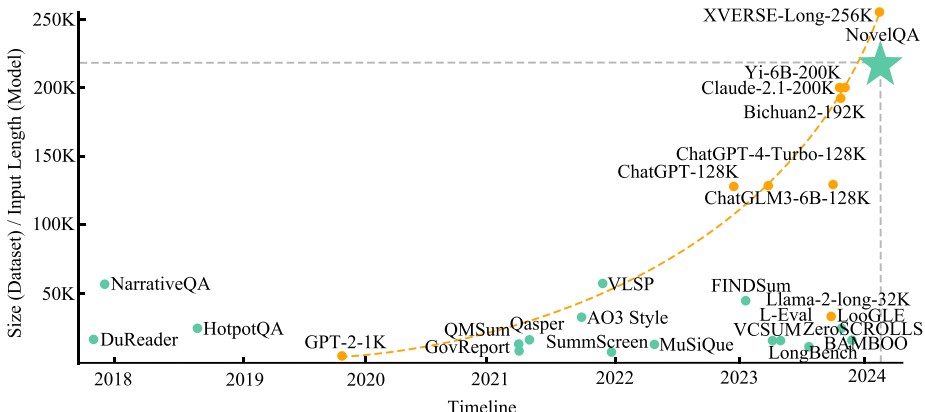

Figure 1: Trend of context window size of LLMs (Orange) and *average* token length of long-range benchmarks (Green). NovelQA is highlighted with a star.

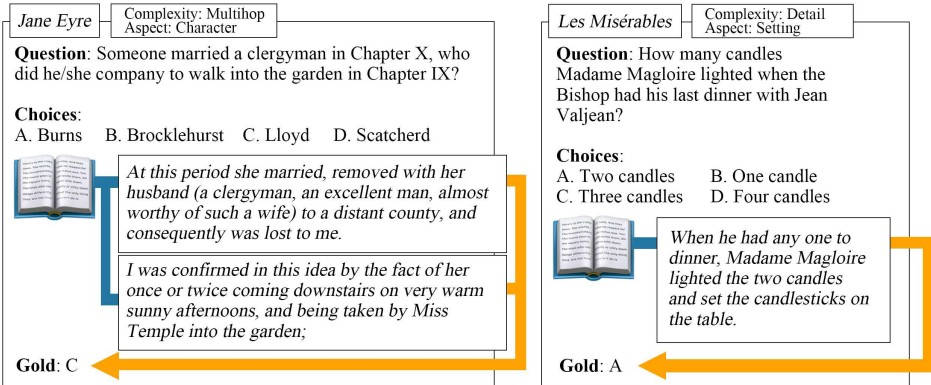

Figure 2: Illustrative examples from NovelQA: This figure showcases two sample questions. For each question, models are evaluated under two distinct settings – multichoice, where the task is to select the correct answer from four options, and Generative, where the model generates an answer.

benchmarks (Shaham et al., 2023; An et al., 2023; Adams et al., 2024), NovelQA addresses the need for assessing extremely long-context understanding, offering a refined and comprehensive tool for advancing natural language processing capabilities. We construct NovelQA based on novels in English, which are also ideal for testing long-context modeling because they are long and complex, with plots that are closely linked from start to end. We select novels from various eras, genres, and formats to enhance diversity. The annotation process is performed by a group of expert annotators, all of whom are holding or pursuing a degree in English Literature and have a strong interest in and familiarity with the novels they annotate. Each question is paired with a 'golden answer' and corresponding textual evidences from the novels. And we categorize them by complexity and aspect for detailed analysis. Figure 2 presents two examples, while Table 1 details the distribution of question types. The dataset includes multi-hop, single-hop, and detail questions, which test the model's abilities to retrieve and integrate scattered information, retrieve information and summarize, and precisely identify specific and subtle details, respectively. The above capabilities are more challenging in a particularly long context. For example, finding relevant information is a particularly big challenge.

We assess various long-context LLMs using NovelQA, including commercial models such as GPT-4-128K (OpenAI, 2023a) and Claude-2.1-200K, Claude-3-200K, Claude-3.5-200K(Anthropic, 2023), alongside open-source models like InternLM2-Chat (Team, 2023) and Llama-3.1 (Touvron et al., 2023). Results show that even the most advanced long-context LLMs face challenges in consistently extracting and processing accurate information from extended texts. For example, Claude-3.5, the top performer, achieves a 62.30% accuracy rate, whereas the open-source Llama-3.1 achieves 51.50% in a generative setting. In addition, some models are unable to answer properly when the context

exceeds 100K tokens, even if their context windows are much longer (xverse, 2023; 01-ai, 2023). Technically, operating LLMs on inputs exceeding 200,000 tokens presents challenges, particularly regarding memory requirements and associated costs. The difficulty of valid models is particularly apparent in answering multi-hop questions and queries that probe meanings, relationships, spans, and timelines, highlighting a significant gap in the models' long-range comprehension. Moreover, our data shows a decline in performance for evidence situated beyond the 100,000-token mark, including information at the novel's end, diverging from the anticipated *lost-in-middle* phenomenon (Liu et al., 2023). This shift suggests a distinct challenge faced by LLMs when processing texts exceeding 100K tokens in length. These results highlight challenges not only in memory optimization but also in the nuanced comprehension and integration of lengthy texts, indicating a substantial obstacle on the path to truly effective long-context LLMs.

To the best of our knowledge, NovelQA is the first long-context QA benchmark featuring manually crafted questions, golden answers, and evidences, with contexts extending beyond 200,000 tokens. [1]

## 2 RELATED WORK

**Long-Range Benchmarks** Evaluating the ability of long-context Large Language Models has been a hot topic (Kočiský et al., 2018; Tay et al., 2021; Shaham et al., 2023; Pang et al., 2022; Wang et al., 2024b; Yu et al., 2024). When entering the era of Large Language Models, the context window length has been much longer than ever, and the decoder-only LLMs have been the mainstream of language models and they have faced specific problems, such as the Lost-in-middle issue (Liu et al., 2023). Thus, increasing benchmarks are created for evaluating long-context LLMs. Among those benchmarks, several are representatives. The *Needle in A Haystack* test (gkamradt et al., 2023) involves inserting an unrelated sentence into a long context and asking the model to retrieve it. However, this test does not represent a natural language question-answering task that requires composite and complex capabilities for processing long contexts. NarrativeQA (Kočiský et al., 2018) asks annotators from MTurk to read novel summaries to write questions. NarrativeXL (Moskvichev & Mai, 2023) utilizes GPT-3.5 to summarize approximately 150 scenes per book from a collection of 1,500 books, and automatically annotates around one million questions. ZeroSCROLLS (Shaham et al., 2023), LooGLE (Li et al., 2023a), Counting-Stars(Song et al., 2024), and LongICLBench (Li et al., 2024) emphasize the importance of understanding and aggregating information from long texts, presenting challenges that highlight areas for improvement in current models. L-Eval (An et al., 2023) introduces a suite of tasks with human-labeled query-response pairs to assess LLMs' performance in processing long inputs effectively, using advanced metrics for a more accurate evaluation. Furthermore, benchmarks like LongBench (Bai et al., 2023), BAMBOO (Dong et al., 2023a), and LongBench-Chat (Bai et al., 2024) offer a diverse set of tasks across languages and domains, from reasoning and coding to summarization and multilingual translation. These benchmarks are designed to rigorously test the ability of LLMs to manage extensive contexts, with LongBench-Chat specifically focusing on instruction-following capabilities in long-context interactions. Additionally, LongHealth (Adams et al., 2024) addresses the need for LLMs to interpret long clinical documents accurately, providing a specialized benchmark for evaluating models on medical texts. This focus on domain-specific challenges underscores the broader necessity for LLMs to not only handle long texts but to do so in a manner that is accurate and contextually relevant across various fields.

NovelQA distinguishes from existing benchmarks in three points. Firstly, NovelQA features an average token length exceeding 200,000, far surpassing the tens of thousands typically found in other benchmarks. Secondly, while other datasets often rely on AI-generation or existing datasets for content creation, the questions, golden answers, and evidences of NovelQA are entirely crafted through human effort of experts. We present a detailed comparison between NovelQA and other benchmarks in Appendix Fig 5.

---

[1]We have released the demonstrations and input of NovelQA, and created a leaderboard. More details can be found in `https://novelqa.github.io/`. And NovelQA is released under the Apache-2.0 License. For the public access, we have released all constructed data on Huggingface `https://huggingface.co/datasets/NovelQA/NovelQA` and an evaluation system on Codabench `https://www.codabench.org/competitions/2727/`. However, we only release public domain novels. Therefore, we offer two types of metrics in the evaluation system and leaderboard: one for evaluating QAs within public domain novels and another for evaluating QAs across all novels.

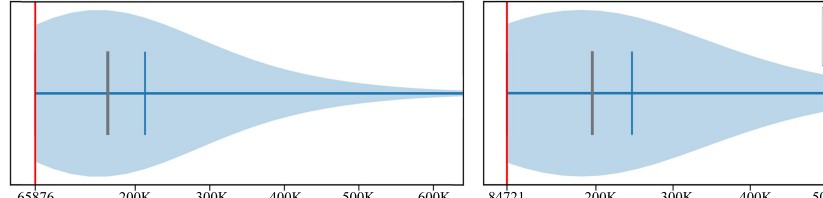

Figure 3: Token Count Distribution in NovelQA, including Copyrighted (left) and Public Domain Only (right). The token counts of both the novel and the questions are counted. The tokenization procedure is accomplished through gpt-3.5-turbo-16k tokenizer.

Table 1: Distribution of Question Types in NovelQA: This table provides a breakdown of questions across different complexity categories (Multi-hop, Single-hop, Detail) and aspect categories (Times, Meaning, Span, Setting, Relation, Character, Plot).

|  | Multi-hop | Single-hop | Detail | Sum |
|---|---|---|---|---|
| **Times** | 463 | 0 | 0 | 463 |
| **Meaning** | 34 | 126 | 206 | 366 |
| **Span** | 34 | 0 | 0 | 34 |
| **Setting** | 24 | 177 | 63 | 264 |
| **Relation** | 119 | 14 | 32 | 165 |
| **Character** | 69 | 255 | 98 | 422 |
| **Plot** | 64 | 414 | 113 | 591 |
| **Sum** | 807 | 986 | 512 | 2305 |

We also discuss related Long-Context Language Modeling methods in Appendix Sec C.1.

## 3 DATA

### 3.1 DATASET OVERVIEW

**Data Formulation** Every novel ($N$) in the dataset corresponds to multiple pieces of annotated data ($d_i$). Each piece of data consists of the following domains, *question* ($Q_i$), *answer* ($A_i$), *multichoices* ($a_{i,0}$, $a_{i,1}$, $a_{i,2}$, and $a_{i,3}$), *gold label* ($a_{i,gold}$), *evidences* ($s_{i,0}$, $s_{i,1}$, ...), and *type* ($Complx_i$ and $Aspect_i$). NovelQA can serve for either a generative or a multichoice task. In the generative setting, a novel text $N$ and a question $Q_i$ are combined to send into the model each time, and the generated answer is compared with the answer $A_i$. In the multichoice setting, the novel $N$, a question $Q_i$, and the four choices $a_{i,0}$ to $a_{i,3}$ are sent into the model, and the output is evaluated according to the gold label $a_{i,gold}$. The *evidences* domain consists of either the original excerpts from the novel or the reasoning steps written by the annotator.

**Book Selection** The books in NovelQA contain 65 free public-domain books from the Project Gutenberg[2] and 24 copyright-protected books purchased from the Internet. The selection of the books follows a criterion to ensure the diversity in the eras, styles, and themes, thus we decided to include a portion of copyrighted novels. All selected books are in English and exceed 50K words (approximately 67k tokens). The distribution of the token count is illustrated in Figure 3.

**Question Distribution** The annotated questions can be classified by the complexity of solving the question and the aspect that the question focuses on. By complexity, the data are categorized into three complexity levels, multi-hop (35.0%), single-hop (42.8%), and detail (22.2%). The order of complexity is as follows: multi-hop > detail > single-hop. By the aspect that each question focuses on, the data entails seven types. A detailed specification of each type is listed in Appendix C.2.2. We have a total of 2305 questions, of which 1640 are from 65 public domain novels, while the remaining 665 are from 24 copyrighted novels. According to the classification above, the distribution of the

---
[2]https://www.gutenberg.org/

questions in our dataset is displayed in Table 1. We also list the ability tested by each kind of question in Appendix Table 10.

**Advantages** Our NovelQA dataset serves as a new benchmark for evaluating long-context understanding, distinguished by several key advantages. Firstly, it surpasses existing benchmarks in length, offering a rigorous test of a model's ability to navigate and comprehend significantly longer texts. Secondly, the inclusion of clear evidences alongside questions ensures that evaluations are grounded in concrete textual support, enhancing the reliability of assessments. Furthermore, the dataset emphasizes questions that require attention to detailed information, challenging models to move beyond superficial impressions to extract specific, nuanced answers. Questions, golden answers, and evidences of the dataset are entirely manually annotated and carefully checked, ensuring high-quality, nuanced questions and answers that reflect complex human thought processes. To prevent against data leakage, we will not release golden answers for the test set, minimizing the risk of overfitting. These features, combined with the dataset's comprehensive coverage of diverse narratives and meticulous construction, make NovelQA a valuable resource for advancing long-context understanding.

## 3.2 DATA ANNOTATION

**Procedure Overview** The annotation process is performed by a group of expert annotators. The annotation procedure consists of two phases: (1) Template-based phase: The annotators can fill entities into 10+ templates (see Appendix Sec C.2.1) that we design to be related to multi-hop or detailed information. This phase entails half of the data, mainly contributing to the multi-hop ones. (2) Free-form phrase: To ensure the diversity of question expression and align the questions to the natural distribution, the second half of our data is annotated without a template, namely, the annotators contribute to any difficult questions that they come up with freely.

**Annotator Recruitment** Our annotators are predominantly English Language and Literature university students or those with a keen interest in English novels, recruited from local universities. These students are accustomed to closely reading English novels for coursework or as a hobby, and writing reports or papers on them. Before annotation, each annotator was instructed to read the annotation instructions to understand the requirements and sign to agree to participate. Annotators are allowed to select novels for annotation based on their familiarity, ensuring they have previously read and comprehensively understood the texts. Meanwhile, we make sure the selected books meet our standards of enough word count and well-developed narratives. We also ensure that each selected novel is either annotated by only one individual, or consistent in version across annotators, despite minor variations among different editions. Each annotator contributes to a typically small number of 20-30 questions per novel. This approach avoids forcing annotators to annotate questions on unfamiliar content.

**Annotation Guideline** While creating the QA tuples, our annotators are instructed to follow several principles below: (1) The annotators are either senior English language and literature college students, or students with high English test scores. (2) The annotators are required to read through the GPT-4 responses on example questions. (3) The annotators should choose novels above 50K tokens and read through the books they choose before annotation. (4) Evidence of each answer should be provided for validation purposes. Evidence should be as sufficient as possible.

**Template Design** The first annotation phase relies on a question template, which requires the annotator to fill in the entities from the novel to form valid questions. To design templates, we carried out sufficient pre-tests on several LLMs to analyze their possible weaknesses in long-input QA and novel knowledge memorization. Our pre-test shows that they usually fail to tackle information spanning over multiple chapters, as well as lack attention to details that have no contribution to the main theme. We also refer to around fifteen books on novel and narration theories (e.g. Forster, 1927; Tobias, 2012; Schmidt, 2012; McKee, 2005) to ensure our template covers more aspects that a novel can discuss (e.g., character, setting, theme). Templates are ensured to test on facts (e.g., events, entities, numbers) that can be traced back to specific evidences from the books, instead of on any subjective feelings or analysis of the readers.

**Time Consumption and Rewards** Given the annotator's familiarity with their chosen novels and their experience with similar questions in their academic assignments, creating questions based on their knowledge becomes a manageable task within a reasonable time cost. The annotation reward is of $1.11 to $1.39 per tuple. As an average annotator can write 5 to 6 pieces of data at full speed

Table 2: Evaluation of Long-Context LLMs on NovelQA. This table presents the performance of four long-context LLMs, including both commercial models (GPT-4, GPT-4o-mini, Claude-2.1, Claude-3-Sonnet and Claude-3.5-Sonnet) and open-source, locally deployed models (InternLM2-Chat-7b/20b and Llama-3.1-8b/70b). Accuracy percentages are reported under two testing scenarios: multichoice and generative. The Max Length column denotes the maximum token length of each model.

|  | Max Length | Multichoice | Generative |
|---|---|---|---|
| GPT-4 | 128K | 71.80 | 46.88 |
| GPT-4o-mini | 128K | 71.85 | 53.32 |
| Claude-2.1 | 200K | 66.84 | 46.04 |
| Claude-3 | 200K | 71.11 | 53.66 |
| Claude-3.5-Sonnet | 200K | **77.92** | **62.30** |
| InternLM2-Chat-7b | 200K | 43.51 | 30.90 |
| InternLM2-Chat-20b | 200K | 49.18 | 32.37 |
| Llama-3.1-8B | 128K | 62.31 | 42.65 |
| Llama-3.1-70B | 128K | **69.39** | **51.50** |
| Human baseline | $\infty$ | **97.00** | **90.00** |

according to our observation, the \$5.56 to \$8.34 hourly wage is above the local legal minimum wage of \$2.78/hour. The annotation process costs around \$3,500.

**Quality Control** The created data is manually double-checked by three authors of this work. The review procedure follows the criteria of minimizing factual errors, enhancing expression clarity, and maintaining challenges to LLMs. Besides, we ensured that all questions are based on factual descriptions and eliminated any subjective ones. Consequently, only 79.4% of the collected data are preserved, resulting in a final dataset of 2305 QA tuples. Meanwhile, we have also conducted the inter-annotator agreement (IAA) test, focusing on evaluating the quality of annotated question-answer pairs. Annotators are required to choose books they are familiar with but have not annotated themselves to answer questions on. As annotators are mostly from the same local university and share similar courses and projects, many can find books they have read in common. We select books read by at least two annotators and have the other reader answer multichoice questions. As the respondents are quite familiar with the target novels and have a strong academic background in English Literature, it takes them around 2-3 hours or less to complete each novel. The IAA test shows a score of 94.6% in Cohen's Kappa, indicating a high agreement among annotators.

**Distractions for Multichoice Setting** We use GPT-4 to generate three distracting options for each question and its golden answer, and randomly permute the four answers.[3] We manually checked those distractions and rewrote those with similar meaning with the golden answers when we double check the data.

# 4 EXPERIMENTS

We focus on long-context models meeting three criteria: a context window of at least 128,000 tokens, accessibility via a full API or public release, and chat functionality. For commercial models, our selection includes GPT-4-128K (OpenAI, 2023a) and Claude 2.1-200K (Anthropic, 2023). Among open-source options, we evaluated models like InternLM2-chat (Team, 2023).

## 4.1 IMPLEMENTATIONS

**Settings** We employ two evaluation settings: a generative setting where models directly generate short answers, and a multichoice setting with four provided options.

**Prompts** We use uniform prompts for all LLMs, with a start part, novel content, questions, choices in the multichoice setting, and an end part. The prompt structure is shown in Appendix Table 11.

---

[3]Our pilot study compared distractors generated by both GPT-4 and Claude 2.1. Interestingly, GPT-4 generated slightly more challenging distractors - both models scored approximately 0.5% lower on GPT-4's distractors compared to Claude 2.1's. This indicates no advantageous bias for GPT-4.

Table 3: Model Performance by Question Type in Generative Setting: This table details the accuracy scores of four models across different question types. Question types include details (dtl), multi-hop (mh), single-hop (sh), character (chara), meaning (mean), plot, relation (relat), setting (settg), others, and an average score (avg) for each category, with '-' indicating the absence of data for a category.

| | (a) GPT-4 | | | | | | | | (b) Claude 2.1 | | | | | | | |
| | chara | mean | plot | relat | settg | span | times | avg | chara | mean | plot | relat | settg | span | times | avg |
|---|---|---|---|---|---|---|---|---|---|---|---|---|---|---|---|---|
| mh | 57.81 | 61.76 | 52.46 | 45.30 | 56.52 | 18.18 | 21.23 | 32.83 | 48.94 | 64.29 | 58.18 | 40.00 | 82.61 | 18.52 | 17.10 | 30.34 |
| sh | 66.12 | 56.56 | 69.33 | 21.43 | 57.23 | - | - | 63.93 | 72.41 | 55.24 | 67.96 | 23.08 | 59.60 | - | - | 65.47 |
| dtl | 52.63 | 12.87 | 61.68 | 37.50 | 58.06 | - | - | 37.58 | 55.22 | 12.43 | 66.30 | 27.59 | 55.77 | - | - | 37.65 |
| avg | 62.04 | 32.40 | 65.93 | 41.72 | 57.38 | 18.18 | 21.23 | 46.88 | 65.90 | 31.61 | 66.60 | 35.61 | 61.06 | 18.52 | 17.10 | 46.04 |
| | (c) InternLM2-Chat-7b | | | | | | | | (d) InternLM2-Chat-20b | | | | | | | |
| | chara | mean | plot | relat | settg | span | times | avg | chara | mean | plot | relat | settg | span | times | avg |
| mh | 23.81 | 38.24 | 42.62 | 24.35 | 39.13 | 15.15 | 21.32 | 24.62 | 32.84 | 44.12 | 29.03 | 37.61 | 25.00 | 15.15 | 26.81 | 29.29 |
| sh | 35.80 | 28.10 | 42.18 | 14.29 | 32.10 | - | - | 36.42 | 42.97 | 34.17 | 43.22 | 21.42 | 36.90 | - | - | 40.57 |
| dtl | 26.67 | 9.18 | 55.14 | 29.03 | 34.43 | - | - | 27.02 | 30.93 | 7.00 | 48.21 | 25.00 | 29.03 | - | - | 24.65 |
| avg | 32.02 | 18.52 | 44.77 | 24.38 | 33.33 | 15.15 | 21.32 | 30.90 | 38.50 | 19.77 | 42.66 | 33.74 | 33.86 | 15.15 | 26.81 | 33.07 |

**Truncation** Due to input length limitations, we truncated the novel content from the end to the front following Bai et al. (2023); Li et al. (2023a); An et al. (2023), to meet with the max input length, while keeping questions and other prompts complete.

**Evaluating Generative Results** Previous researches have proved the ability of LLMs in evaluating the machine-generated answers align with human judgements (Wang et al., 2023; An et al., 2023; Li et al., 2023a) After a pilot study showing that the models show no several preference towards the answers generated by its own kinds (See Appendix C.4.2, we choose GPT-4 to evaluate our generated results. We further conducted a human evaluation on 800 pieces of generative outputs and carried out an inter-evaluator agreement (IEA) test between two human evaluators and the GPT-4 evaluator. In the IEA test, annotators also serve as human evaluators of the novels they were familiar with. Their evaluations were compared to those of GPT-4 evaluators, with Cohen's kappa score calculated to measure agreement. As shown in Table 7, the result of 89.25% in Cohen's Kappa indicates a high agreement towards the GPT-4 evaluating results. NovelQA primarily consists of objective questions, which have clear, verifiable answers with a factual basis in the text. This objectivity significantly reduces the impact of model bias in LLM-as-Judge evaluation

**Commercial LLMs** The APIs of commercial LLMs utilized are gpt-4-0125-preview, gpt-4o-mini, Claude-2.1, Claude-3-sonnet and Claude-3.5-sonnet.

**Open-source LLMs** Running long-context LLMs on extremely long inputs, such as 200K tokens, is a challenge due to the immense GPU memory required, for example, it takes roughly 2.5T memory to calculate one attention matrix for a 7B model with a 200K-token input, while our local device is a $4 \times 80G$ A100. To address this, we utilize the LMDeploy (Contributors, 2023) (based on Dynamic NTK (emozilla, 2023)) and vLLM (Kwon et al., 2023) for memory and time reduction, which is only compatible with several LLMs. Therefore, we choose InternLM2-Chat-7b-200K, InternLM2-Chat-20b-200K, Llama-3.1-8b, and Llama-3.1-70b for our experiments.

**Human Performance** We selected books that have been read by multiple annotators, and then have had the readers engage in a two-round answering process on novels they had not previously annotated. The first round was in a generative setting, and the second round was in a multichoice setting. This process was conducted on 5 novels with a total of 100 questions. The result shows that human performance scored 90 in the generative setting and 97 in the multiple-choice setting.

## 4.2 MAIN RESULTS

We present the main results in Table 2. Even the highest scores (71.80% and 46.88% for GPT-4 in generative and multichoice settings, respectively) suggest there is considerable room for improvement in long-context understanding compared with human readers. This is especially true in the generative setting where understanding and recall over long contexts are more challenging. Additionally, commercial models (Claude-3.5-Sonnet and GPT-4o-mini) outperform open-source models in this benchmark. Among closed-source models, LLaMA family models outperform the InternLM models. All models show a drop in performance in the generative setting compared to the multichoice setting.

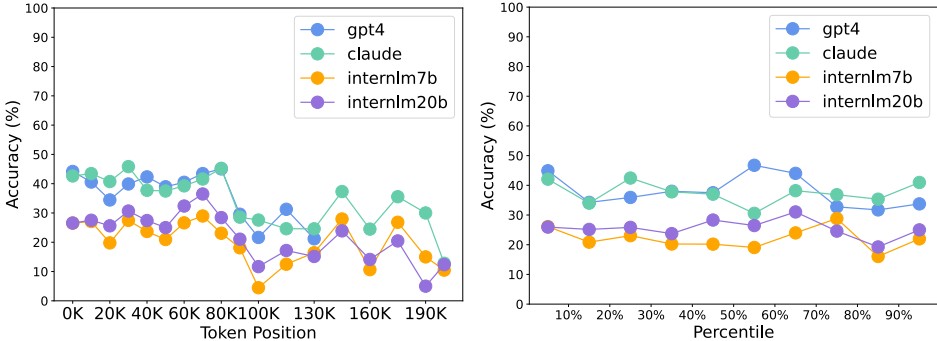

Figure 4: Analysis of Accuracy in Generative Setting by Absolute and Relative Token Positions: The two figures above illustrate the accuracy, plotted against the absolute token indexes (left) and the percentage position (right) of each question's relevant evidences. The x-axis of the absolute token position figure (left), reflecting token indexes, is folded on the right due to the long-tails.

This indicates the increased challenge in generating a correct answer from scratch, as opposed to selecting from provided options.

We appended a typical error analysis in Appendix Sec C.4.6.

### 4.3 RESULTS BY THE QUESTION TYPE

An in-depth analysis of model performance across question types reveals nuanced insights into their comprehension abilities in both generative and multichoice settings, detailed in Table 3 and Appendix C.4.3, respectively. This analysis not only highlights the models' weaknesses across different formats but also illuminates the challenges in narrative comprehension, contributing to both NLP and computational literary research.

The examination of accuracy scores across question categories such as character, meaning, plot, relation, and setting highlights distinct patterns in performance, pointing to the models' differential capabilities and limitations. Notably, models exhibit particular difficulty with questions centered around meaning, relation, span, and times. This difficulty suggests several underlying challenges:

(1) **Meaning Questions**: The struggle with meaning questions indicates a challenge in grasping abstract concepts and locating entities or sentences through interpretations within the text, which requires an advanced level of semantic understanding and inference.

(2) **Relation Questions**: Difficulty with relation questions points to a gap in the models' ability to identify and interpret the dynamic and often nuanced relationships between characters, events, or concepts, which are crucial for a holistic understanding of narratives.

(3) **Span and Times Questions**: The lower performance on span and times questions suggests a limitation in tracking temporal sequences and spatial extents within the narrative, reflecting challenges in maintaining and applying contextual information over long stretches of text.

The above findings underscore a critical aspect in both computational literary studies and long-context comprehension of LLMs—while models are adept at handling certain types of narrative questions, they encounter significant hurdles when required to synthesize abstract concepts, interpret complex relationships, or maintain a coherent understanding of temporal and spatial narratives over long context. These can be the domains requiring further improvement to enhance narrative comprehension and reasoning capabilities.

### 4.4 RESULTS BY POSITION

Our analysis delves into how the positioning of evidence within novels affects the accuracy of long-context LLMs. Specifically, we explore the impact of both absolute and relative positions of evidence, where the absolute position refers to the specific token index within the text, and the relative position is normalized against the total length of the novel, scaled to a 0%-100% range.

Table 4: Model Performance Analysis Pre and Post 100K Tokens: The accuracies of questions on evidences pre-100K and post-100K are calculated separately and compared, where the '100K' indicates the aboslute starting token position of evidences relating to answer. The result shows an obvious contrast as the accuracy drops sharply after the 100K, in both generative and multichoice settings on each model.

|  | Multichoice | | Generative | |
| --- | --- | --- | --- | --- |
|  | pre-100K | post-100K | pre-100K | post-100K |
| GPT-4 | 83.17 | 59.22 | 53.64 | 32.01 |
| Claude2.1 | 77.65 | 54.46 | 61.96 | 30.57 |
| InternLM-7b | 51.78 | 38.17 | 37.71 | 18.23 |
| InternLM-20b | 55.59 | 40.87 | 37.74 | 21.10 |
| Weighted Avg. | 66.24 | 48.08 | 46.84 | 25.36 |

Table 5: Evidence recalling results of four LLMs: During the evidence recall test, each model is required to output the relevant evidence from the original novel contents. Each recalled piece of evidence is further compared by GPT-4, which has been proven to generate reliable evaluations, with the gold standard evidence. The scores for each model, assessed in three dimensions, Correctness, Relevance, and Sufficiency, are listed below.

|  | Correctness | Relevance | Sufficiency | Avg. |
| --- | --- | --- | --- | --- |
| GPT-4 | 29.47 | 38.05 | 27.67 | 31.73 |
| Claude 2.1 | 23.51 | 29.08 | 22.27 | 24.95 |
| InternLM2-Chat-7b | 2.05 | 3.88 | 1.90 | 2.61 |
| InternLM2-Chat-20b | 6.36 | 12.51 | 7.50 | 8.79 |

**Absolute Position** In the generative setting, as depicted in Figure 4 (left), all evaluated models show improved performance on questions where the necessary evidence is located before the 100K token mark. This trend highlights a challenge for LLMs in accessing and processing information beyond this threshold, suggesting a diminished capacity to handle very long inputs. The multichoice setting, detailed in Appendix C.4.4, follows a similar pattern, reinforcing the importance of evidence position in model performance. We also present the relationship between the accuracy and the absolute position of evidence, grouping by pre-100K and post-100K, in Table 4.

**Relative Position** By normalizing the evidence positions within the entire novel, we aim to understand if the proportional location of evidence influences model accuracy. This analysis, shown in Figure 4 (right) for the generative setting and in the Appendix C.4.4 for the multichoice setting, indicates that models maintain relatively consistent performance across various relative positions, suggesting that long-context LLMs' effectiveness is not significantly affected by the evidence's relative position within the standardized length of novels.

To delve deeper into how long-context LLMs navigate extremely long inputs, we segment novels into two categories based on length: 65k-100K and over 100K tokens. We then examine model accuracy in relation to relative evidence positions within these ranges, with results presented in Appendix Figure 7. For novels within the 65k-100K token range, we observe a *lost-in-middle* phenomenon in GPT-4, InternLM2-7b, and InternLM2-20b, akin to findings by Liu et al. (2023). This pattern indicates stronger performance at the beginning and end of texts but weaker in the middle. Conversely, in novels exceeding 100K tokens, model performance generally declines towards the end, potentially due to the scarcity of training data for contexts of this length. This behavior underscores a unique challenge faced by LLMs when processing exceptionally long texts over 100K tokens.

This analysis highlights the critical role of absolute evidence positioning in determining the accuracy of LLMs in processing long texts. Challenges arise when context beyond a specific token threshold. Conversely, the relative position within a normalized length has a minimal effect.

## 4.5 EVIDENCES RECALL

We evaluate the ability to recall evidence, prompting the four models above to answer the questions in NovelQA again, with printing the supporting evidence simultaneously. We then prompt GPT-4

Table 6: Close-book Performance across four LLMs on NovelQA. Unlike the standard scenario, models rely solely on internal knowledge without access to the novels. The parentheses indicate the performance drop from the standard to the Close-book scenario, highlighting the models' dependency on external text for answering.

| | Close-book QA | |
| --- | --- | --- |
| | **Multichoice** | **Generative** |
| GPT-4 | 60.94 (-9.06) | 34.30 (-13.58) |
| Claude 2.1 | 51.77 (-15.01) | 22.36 (-20.68) |
| InternLM2-Chat-7b | 33.58 (-10.71) | 14.12 (-16.81) |
| InternLM2-Chat-20b | 33.05 (-16.13) | 15.51 (-16.86) |

with the generated evidence alongside the annotated evidence to obtain its evaluation on the quality of retrieved evidence pieces. The evaluating matrix consists of the following three dimensions: *correctness* refers to whether the retrieved evidence is the same as the annotated evidence or with a similar correct meaning; *relevance* indicates whether the evidence is consistent with the answer; *sufficiency*, whether the retrieved pieces of evidence are enough to support the answer. Each dimension is scored between 0 and 100 and an average score is further obtained through calculating the algorithmic mean on these three dimensions. Prompts involved in this evaluation procedure are presented in Appendix C.2.3.

The results, detailed in Table 5, show higher performances of GPT-4 and Claude 2.1. Moreover, though the scoring range is from 0 to 100, the four models all perform with low scores in evidence recall, possibly because the models do not always follow the instructions after inputs and outputs of such an above-average length, resulting in a high percentage of 0 scores. This phenomenon is particularly severe for InternLM-Chat-7b and InternLM-Chat-20b models, whose outputs consist of a large proportion of invalid placeholders. Still, current long-context LLMs generally demonstrate inadequate abilities recalling the correct and supportive evidences from the context.

### 4.6 CLOSE-BOOK QUESTION ANSWERING

We employ a Close-book QA scenario to assess the extent to which models rely on the content of novels versus using their internal knowledge to answer questions. In this approach, the models are not given access to the text of the novels and must rely solely on their pre-existing knowledge to provide answers. Given that our selected novels are well-known and representative of their genres, it's inevitable that LLMs have encountered their texts during training and retained some of their content. The results of this evaluation are presented in Table 6. Models like GPT-4 and Claude 2.1 achieve notable scores in the Close-book setting (60.94% and 51.77% in multichoice, 34.30% and 22.36% in generative, respectively) indicating that they have internalized significant portions of the novels' content during training. This internal knowledge allows them to perform reasonably well even without direct access to the text. The difference in performance between the Close-book and standard settings underscores the challenges in long-context understanding. While models can retain and recall information from well-known texts, their ability to comprehend and use such information to answer questions accurately diminishes in the absence of the text. This suggests that long-context understanding, as measured by the main results, might still be more challenging than it appears, as models benefit from having the text directly available.

### 5 CONCLUSION

We introduced NovelQA, a long-range benchmark designed to assess the long-context comprehension abilities of LLMs. Utilizing representative English novels, NovelQA presents LLMs with the challenge of navigating complex, real-world texts. Our evaluations reveal that both commercial and open-source models face challenges with detailed understanding, multi-hop reasoning, and accurately retrieving specific information from lengthy contexts, especially for lengths are 100,000. Moreover, operating LLMs on inputs exceeding 200,000 tokens faces technical challenges, notably in terms of memory requirements and associated costs.

ACKNOWLEDGMENT

We appreciate the contributions of our annotators: Jingyi Wu, Ziyue Li, Qiangyi Zhu, Xinyao Feng, Zhangrui Xu, Xiao Wu, Yu Qiu, Hanxin Ye, Mengting Wei, Hanyu Cai, and Yaokai Tu from Hangzhou Normal University; Ziyang Zhu, and Xinyu Zhu from Ningbo University; and Rizwan Abbas, Khadija Azhar, and Snawar Hussainfrom Zhejiang University. This publication has emanated from research conducted with the financial support of the National Natural Science Foundation of China (NSFC No.62161160339)

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

## A  LIMITATIONS

**Genre Limitation**: NovelQA focus on the novel genre. The contents of several popular novels might be mentioned in existing datasets, resulting in the risk of containing the correct answers of our copyright protected books, although this dataset has managed to contain non-trivial details and reasoning questions to avoid this risk.

**Language Limitation**: NovelQA, and all associated data are exclusively in English. The following researches may consider to extend the language covered.

**Truncation**: Truncation is a standard practice for adapting longer texts to fit within the constrained context windows of LLMs, especially when evaluating their performance on long-context tasks. While it's possible that some evidence may fall within the truncated parts.

## B  ETHICS STATEMENTS

We are dedicated to ensuring that NovelQA serves exclusively for academic and scientific endeavors. Our plan includes the launch of an evaluation website, a leaderboard website, and the provision of an API for data access. As certain novels used in our project are protected by copyright, we affirm that we will not release these novels.

NovelQA does not contain any personally identifiable information or offensive content.

## C  APPENDIX

### C.1  RELATED WORKS

**Long-Context Language Modeling** can be divided into several parts including efficient attention, preserving long-term memory like using KV cache, extrapolative positional embedding module, context pre/post-processing (Huang et al., 2023; Dong et al., 2023b). Using efficient attention can reduce computational complexity and memory usage (Beltagy et al., 2020; Wang et al., 2020; Tworkowski et al., 2023; Li et al., 2023b). Preserving KV cache or context-level cache allows models to recall and leverage past information without reprocessing, enhancing coherence over long texts (Chevalier et al., 2023; Wu et al., 2022a;b; Lin et al., 2024; Zhong et al., 2023; Chen et al., 2023a; Hooper et al., 2024). Deploying an extrapolative positional embedding can extend beyond the sequence lengths seen during training (Chen et al., 2023b; Su et al., 2024). Pre/post-processing the context can make models focus on key information (Li et al., 2023c; Jiang et al., 2023; Zhang et al., 2024; Han et al., 2023). Moreover, some training methods are put forward to support further long-context language modeling development (Wang et al., 2024a; Press et al., 2021). Since these models are more about exploring long-context language modeling without scaling in data and model size, the focus is on language modeling indicators, such as BLEU and Perplexity, NovelQA can serve as a benchmark for evaluating those long-context language modeling methods.

### C.2  DATA

Table 7: The Cohen's Kappa score in Inter-Evaluator Agreement test on different model outputs between human-evaluator and GPT4-as-evaluator. Higher Cohen's Kappa scores indicate higher agreement between human-evaluators and the GPT-4-evaluator.

| Model | To Human Evaluator A | To Human Evaluator B | Avg. |
|---|---|---|---|
| GPT-4 | 91.97 | 95.84 | 93.91 |
| Claude 2.1 | 88.04 | 90.00 | 89.02 |
| InternLM-Chat-7b | 91.88 | 86.53 | 84.88 |
| InternLM-Chat-20b | 85.68 | 84.08 | 87.25 |
| Avg. | 89.39 | 89.11 | 89.25 |

| | Avg. Length Over 200K | Long Dependency | Details | Require Reasoning | Multiple Tasks |
|---|---|---|---|---|---|
| BAMBOO | ❌ | ✅ | ❌ | ❌ | ✅ |
| LongBench | ❌ | ✅ | ❌ | ❌ | ✅ |
| ZeroSCROLLS | ❌ | ❌ | ❌ | ✅ | ✅ |
| LongICLBench | ❌ | ✅ | ❌ | ✅ | ❌ |
| Counting Starts | ❌ | ✅ | ✅ | ❌ | ❌ |
| L-Eval | ❌ | ❌ | ✅ | ✅ | ✅ |
| LooGLE | ❌ | ✅ | ✅ | ✅ | ✅ |
| **NovelQA** | ✅ | ✅ | ✅ | ✅ | ✅ |

Figure 5: Comparison between NovelQA and other long-context benchmarks highlights several key dimensions. *Long dependency* refers to questions that require multi-hop evidence spanning extensive text to resolve. *Details* pertains to questions that need answers with minimal contextual support. *Reasoning* involves questions without direct answers in the text, necessitating inference. NovelQA excels in these dimensions and features the longest context length among the benchmarks, catering to the current situation where a number of LLMs have a context length equal to or over 200K.

Table 8: Selected question templates adopted in data annotation. Question types include character (chara), meaning (mean), plot, relation (relat), setting (settg), times and span. '<>' label indicates the entity for annotators to fill in. Note that the annotators are also encouraged annotate questions beyond the given templates.

| Aspect | Template | Answer Format |
|---|---|---|
| times | Has the plot <a plot> happened in the novel? If so, how many times does it happen in the text? | Yes/No + number |
| | How many times has <a character> done <a doing-verb phrase> in the novel? | Yes/No + number |
| | How many times have <a character> and <a character> <do something> together in the novel? | Yes/No + number |
| | How many times have <a character> and <a character> <met each other (or appear together)> in the novel? | Yes/No + number |
| | How many times have <a character> and <a character> <communicate or have verbal conflicts with each other> in the novel? | Yes/No + number |
| mean | Explain the meaning or implication of the symbol or metaphor <a symbol or a metaphor> in one sentence, which appears in the novel. | An explanation |
| | In which chapter does there exist a sentence in the novel <the novel> with the same or similar meaning as "<a sentence not from the original text>"? Please output the chapter name or index. | A chapter index or title |
| chara | <A character> is used to be <positive or negative> and finally becomes a <negative or positive> one in the novel. Tell in one sentence which episode marks this character's change. | A plot in one sentence |
| | Who are mentioned with names in the <an organization, a family, or a club> in the novel? | A list of names |
| | Please list 3 aliases or designations of <a character> in the novel. | 3 aliases or designations |
| | Who is <a minor character, or a character without name, or a character that appears only once> in the novel? | A description of character |
| settg | In which <cities, or countries> does this story take place in the novel? | A list of cities or countries |
| | In which year does the earliest event happen, and in which year does the latest even happen in the novel? | A range in years |
| relat | What is the relationship between <a character> and <an alias or a nickname of this character> in the novel? | A relationship |

### C.2.1 QUESTION TEMPLATES

Question templates adopted in the annotation procedure are presented in Table 8. The templates are designed using a combination of literary theories and widely-used long-context tests on LLMs. During annotation, annotators are required to fill in the missing entities (e.g., character, place, sentence) in the templates and are allowed to modify the expressions to suit specific questions. This approach balances the workload of annotation and ensures a diverse set of questions.

## C.2.2 DATA CLASSIFICATION

Table 9: Data Distribution of NovelQA. Each question is labeled with a Complexity and an Aspect label. The Complexity dimension comprises three categories, multi-hop, single-hop, and detail, while the Aspect dimension comprises seven categories including times, meaning, and so forth.

| Dimension | Type | Percentage | Description | Example |
|---|---|---|---|---|
| By Complexity | Multi-hop | 34.98% | Questions requiring knowledge across multiple paragraphs, or even multiple chapters, to be solved. | *Has Crusoe been on a voyage? If so, how many times has this plot happened?* in *The Life and Adventures of Robinson Crusoe* |
| | Single-hop | 42.74% | Questions requiring knowledge from one or several adjacent single sentences to be solved. | *According to the Colonel, what did he smoke in McQueen's compartment?* in *Murder on the Orient Express* |
| | Detail | 22.19% | Questions requiring knowledge from one or two adjacent sentences to be solved. Detail questions are distinguished from the single-hop class by involving information that is too minor to impact other plots, making the details difficult to recall. | *How many candles Madame Magloire lighted when the Bishop had his last dinner with Jean Valjean?* in *Les Misérables* |
| By Aspect | Times | 20.07% | About the number of times that a character, location, or plot appears in the novel. | *How many times has Kitty kissed Walter?* in *The Painted Veil* |
| | Meaning | 15.86% | The understanding of certain sentences or metaphors, e.g., to interpret the relationship of a certain metaphor and the novel itself, or find a specific sentence according to a paraphrase provided by the annotator. | *In which chapter does there exist a sentence with the same or similar meaning as 'The Marquis responded, 'You do me too much honor. In any case, I lean toward that assumption."?* in *A Tale of Two Cities* |
| | Span | 1.47% | About the range of the novel setting. To be specific, they either ask about the starting and ending year of the story or require listing all the cities or countries that are involved in the story. | *In which year does the earliest event happen, and in which year does the latest even happen?* in *Tess of the d'Urbervilles* |
| | Setting | 11.44% | About the time or place settings, besides those in the span type, are classified in this type. | *Where did Diana's cousins leave for the Debating Club concert?* in *Anne of Green Gables* |
| | Relation | 7.15% | About the relationship of multiple character entities. To be specific, they ask either about the relationship of a character and their alias or designation, or about the relationship between different characters. | *What is the relationship between Jean Valjean and Ultime Fauchelevent?* (designation) *Who are members of ABC friends?* (membership) in *Les Misérables* |
| | Character | 18.29% | About the information of characters, besides those in the relation type, are classified into this type. | *Who is Miss Beirne?* in *Dubliners* |
| | Plot | 25.62% | We define a plot as "some character does something for once". Questions that ask about the information of any plots are classified into this type. | *What does Clarissa repair in preparation for the night's party?* in *Mrs.Dallory* |

The data in NovelQA are classified into 3 complexity levels and focus on 7 aspects. Table 9 presents a detailed description of each class's criteria and examples.

## C.2.3 PROMPTS

We present all the prompts involved in both test and evaluation phases in Table 11. The prompts generally follow the formula of zero-shot prompting. In each prompt, the model is firstly assigned the identity of a literature professor reviewing student answers. Then the prompt provides the model with a detailed task description and clear specifications for the input and desired output.

## C.2.4 THE LENGTH DISTRIBUTION OF EVIDENCES

Of the 3,666 evidence instances, 134 are summaries or comments without exact matches in the document. The distribution of the remaining 3,532 instances is as follows: Absolute distribution (by

Table 10: Question types and their corresponding LLM abilities. Each type of question in NovelQA is designed to test one or more abilities of LLMs, including the abilities to retrieve information, to identify details, and so forth.

| Question Type | Abilities Required |
|---|---|
| Multi-hop | Ability to retrieve and integrate scattered information. |
| Single-hop | Ability to retrieve information and summarize. |
| Detail | Ability to precisely identify specific, subtle details. |
| Times | Ability to retrieve facts, integrate facts, and reason. |
| Meaning | Ability to retrieve ambiguous information and reason. |
| Span, Setting, Relation | Ability to retrieve facts, integrate facts, and reason. |
| Character, Plot | Ability to subtle or ambiguous information and reason. |

absolute token index) can be found in Table 12: Approximately 18.4% of the evidence is located beyond the 130K token mark (outside GPT-4's context window), and 6.4% is beyond 210K tokens (outside Claude and InternLM's context windows).

Given the varying lengths of novels, we also analyzed the relative position distribution (by relative token position: token position / total tokens in the corresponding book) in Table 13:

Our analysis reveals that the highest proportion of evidence instances (25.17%) occurs within the first 10% of the novels, while the distribution across the middle sections is relatively uniform. This concentration in the early parts of the novels can be attributed to the initial introduction of characters and plot elements. The first occurrence of evidence related to these introductions naturally falls in the earlier sections of the novels. It's important to note that during the question formulation process, we did not deliberately adjust the distribution of questions. The observed pattern in answer locations emerges naturally from the narrative structure of the novels.

### C.3 MULTICHOICE DISTRACTORS GENERATION

Our pilot study compared distractors generated by both GPT-4 and Claude 2.1. Interestingly, GPT-4 generated slightly more challenging distractors - both models scored approximately only 0.5% lower on GPT-4's distractors compared to Claude 2.1's. Thus, we consider the bias caused by the model choice not significant in the distractors generation, and safely chose the choices generated by GPT-4.

### C.4 EXPERIMENTS AND ANALYSIS

#### C.4.1 IMPLEMENTATIONS

We also tried to test Gemini-1.5[4], but the generation of 816 questions from 29 novels was blocked for unknown reasons (with only 14 of these novels under copyright protection). Consequently, we have decided not to present Gemini's result.

**Experiment Configs** Given the cost of running long-context APIs, we request that each model respond to all questions for a given book in a single session. To ensure fair comparisons, local-deployed LLMs also answer all questions for a book at once. We set 'temperature = 0' to eliminate randomness and keep other hyper-parameters default.

#### C.4.2 EVALUATOR MODEL

To prove that the model bias has little or no effect on our final results, we conducted a thorough analysis comparing different evaluator models to check their potential, demonstrated in Table 15. The results show minimal variance between evaluators, with Claude-3.5 being slightly stricter but showing no significant model preference. This consistency is largely due to NovelQA's objective nature, with well-defined questions and answers that leave little room for evaluator bias.

---

[4]https://deepmind.google/technologies/gemini/flash/

Table 11: Prompts used in each setting of question answering. In each prompt, the model is assigned the identity of a literature professor reviewing student answers, with a detailed task description and clear specifications for the input and desired output. The angle brackets '<>' indicate the contents varying among each input.

| QA Setting | Prompt |
|---|---|
| Generative | You are a literature professor. I will provide you with the full text of a novel along with a series of questions. Please thoroughly analyze the novel's content to accurately respond to each of the following questions. Book title: <title>; Book Content: <content>; Book ends. Questions start here: N ×(Question: <question>); Questions end here. Try your best to answer the questions based on the given novel full text. The answer should be in short with only one or several words. Your output format should be 'Answer0: <answer>Answer1: <answer>... Answern: <answer>', each answer in one line without outputting the questions and other info. |
| MultiChoice | You are a literature professor. I will provide you with the full text of a novel along with a series of questions and corresponding choices pertaining to it. Please thoroughly analyze the novel 's content to accurately respond to each of the following questions. Book title: <title>; Book Content: <content>; Book ends. Questions start here: N ×(Question: <question> Choices: 0: <choice0> 1: <choice1> 2: <choice2> 3: <choice3>); Questions end here. Try your best to select the correct choice to each question based on the given full text the novel. Your should output the choice to each question with the format 'Answer0: <choice> Answer1: <choice>... Answern: <choice>' (only the choice index is required), each answer in one line without outputing the questions and other info. |
| Closebook-Generative | You are a literature professor. I will provide you a series of questions. Please accurately respond to each of the following questions. Book title: <title>; Book Content: <content>; Book ends. Questions start here: N ×(Question: <question>); Questions end here. Try your best to answer the questions based on your own knowledge. The answer should be in short with only one or several words. Your output format should be 'Answer0: <answer>Answer1: <answer>... Answern: <answer>', each answer in one line without outputting the questions and other info. |
| Closebook-MultiChoice | You are a literature professor. I will provide you a series of questions along with four choices for each question. Please accurately select the correct choice to each of the following questions. Book title: <title>; Book Content: <content>; Book ends. Questions start here: N ×(Question: <question> Choices: 0: <choice0> 1: <choice1> 2: <choice2> 3: <choice3>); Questions end here. Try your best to answer the questions based on your own knowledge. Your should output the choice to each question with the format 'Answer0: <choice> Answer1: <choice>... Answern: <choice>' (only the choice index is required), each answer in one line without outputting the questions and other info. |
| Evaluating Generative | You are a literature professor reviewing a student's quiz paper. The question is about the novel <novel title>: <question>. The related evidences from the novel are: <evidences>. Correct ans is: <ca>. Student ans is: <sa>. Plz check whether the student's ans is correct wrt. the correct ans, and return "C" for correct and "N" for not correct. esp., if the student grabs the correct ans's meaning, return "C". However, if there are factuality errors in student ans, or the question requires a specific number but the student answers a rough number, you should return "N". Please only return the char C or N w/o any other output. |
| Evidence Recall | You are a literature professor. I will provide you with the full text of a novel along with a series of questions. Please thoroughly analyze the novel's content to accurately respond to each of the following questions. Book title: <title>; Book Content: <content>; Book ends. Questions start here: N ×(Question: <question>); Questions end here. Try your best to answer the questions based on the given full text of the novel. The answer should be in short with only one or several words. Your output format should be Ánswer0: <answer>$ <evidences> Answer1: <answer>$ <evidences>... Answern: <answer>$ <evidences>¡ each answer in one line with all the supporting evidences. Each evidence should be a sentence exactly from the original text without any paraphrase. |
| Evaluating Evidence Recall | You are a literature professor reviewing student's evidence for their answer about novel <novel title>. Question: <ques>. Correct answer: <ca>. Student answer: <sa>. Correct evidence: <ce>. Student evidence: <se>. You should evaluate the student evidence in 3 aspects: C) correctness: whether the student evidence is the same with the correct evidence or with a similar correct meaning. R) relevance: whether the evidence is relevant to the ans. S) sufficiency: whether sufficient evidences are retrieved to support the ans. And give a score of 1-100 to only the evidence (not the ans). You should **only** return 3 score numbers, e.g.in format C50R66S33, without any other outputs. |

### C.4.3 MULTICHOICE PERFORMANCE RELATED TO QUESTION TYPE

Besides the evaluation in generative settings, we also prompted the models to collect their responses for the multichoice versioned questions. Table 14 presents the accuracies of four models in multichoice settings in each question type.

Table 12: The distribution of evidences by absolute token index.

| Range | 0~20K | 20~40K | 40~60K | 60~80K | 80~100K | 100~130K | 130~170K | 170~210K | >210K |
|---|---|---|---|---|---|---|---|---|---|
| Count | 959 | 631 | 446 | 359 | 249 | 239 | 214 | 209 | 226 |
| Percentage | 27.15 | 17.87 | 12.63 | 10.16 | 7.05 | 6.77 | 6.06 | 5.91 | 6.40 |

Table 13: The distribution of evidences by relative token index.

| Range | 0~10 | 20 | 30 | 40 | 50 | 60 | 70 | 80 | 90 | 100 |
|---|---|---|---|---|---|---|---|---|---|---|
| Count | 889 | 431 | 360 | 326 | 288 | 271 | 274 | 250 | 205 | 238 |
| Percentage | 25.17 | 12.20 | 10.19 | 9.23 | 8.15 | 7.67 | 7.76 | 7.08 | 5.80 | 6.74 |

Table 14: Model Performance by Question Type in Multichoice Setting: This table details the accuracy scores of four models across different question types within the Multichoice setting of NovelQA. Question types include character (chara), meaning (mean), plot, relation (relat), setting (settg), and others, with '-' indicating the absence of data for a category. The table also provides an average score (avg) for each question category and model. Abbreviations used are dtl (details), mh (multi-hop), sh (single-hop).

| (a) GPT-4 | | | | | | | | (b) Claude 2.1 | | | | | | | |
|---|---|---|---|---|---|---|---|---|---|---|---|---|---|---|---|
| | chara | mean | plot | relat | settg | span | times | avg | chara | mean | plot | relat | settg | span | times | avg |
| mh | 55.88 | 44.12 | 53.12 | 56.78 | 66.67 | 44.12 | 38.53 | 45.15 | 71.88 | 76.47 | 85.00 | 70.69 | 78.26 | 51.52 | 47.71 | 58.17 |
| sh | 53.94 | 53.97 | 58.94 | 35.71 | 62.15 | - | - | 57.26 | 82.28 | 76.72 | 83.29 | 50.00 | 80.89 | - | - | 81.19 |
| dtl | 47.96 | 25.98 | 55.75 | 18.75 | 55.56 | - | - | 30.00 | 63.51 | 38.12 | 76.92 | 68.75 | 79.03 | - | - | 58.02 |
| avg | 52.86 | 37.36 | 57.70 | 47.56 | 60.98 | 44.12 | 38.53 | 49.18 | 76.80 | 54.55 | 82.21 | 68.52 | 80.17 | 51.52 | 47.71 | 66.78 |
| (c) InternLM2-Chat-7b | | | | | | | | (d) InternLM2-Chat-20b | | | | | | | |
| | chara | mean | plot | relat | settg | span | times | avg | chara | mean | plot | relat | settg | span | times | avg |
| mh | 42.19 | 38.24 | 45.90 | 46.15 | 65.22 | 39.39 | 42.92 | 43.87 | 76.81 | 88.24 | 87.50 | 79.83 | 91.67 | 52.94 | 45.79 | 60.22 |
| sh | 44.44 | 39.34 | 44.56 | 28.57 | 48.15 | - | - | 44.23 | 86.27 | 88.10 | 92.03 | 57.14 | 87.01 | - | - | 88.64 |
| dtl | 52.63 | 26.24 | 55.14 | 31.25 | 59.68 | - | - | 41.54 | 69.39 | 30.10 | 85.84 | 53.12 | 80.95 | - | - | 57.62 |
| avg | 45.69 | 31.84 | 46.79 | 41.72 | 52.63 | 39.39 | 42.92 | 43.51 | 80.81 | 55.46 | 90.36 | 72.73 | 85.98 | 52.94 | 45.79 | 71.80 |

### C.4.4 MULTICHOICE PERFORMANCE RELATED TO EVIDENCE POSITION

Figure 6 which presents the relationships between the accuracy and the absolute or relative positions accordingly shows similar trends to those observed in the generative setting. To be specific, the accuracy by absolute token position remains high when related evidences are before 100K's text length, while it drops after 100K. Meanwhile, the accuracy by relative position remains relatively even. A comparison is made between the accuracies within two ranges, 65K(the lowest token count) to 100K (namely pre-100K) and 100K to the end (namely post-100K). Figure 7 and Table 4 present a clearer contrast between these two ranges, where the precision drops dramatically after the 100K token.

### C.4.5 MULTI-HOP ACCURACY RELATED TO EVIDENCE DISTANCE

We also measured the relationship between the evidence distance within each multi-hop question and the accuracy under the generative task. For each multi-hop question, we obtain the distances among all evidences, and consider the max distance among them as the evidence distance. This can be interpreted as the model must memorize at least one of its evidences for the max distance to meet the final evidence in order to obtain the answer. The correlation between the evidence distance and the accuracy for multi-hop questions is demonstrated in Table 16. Among the indices, Pearson correlation assumes that the two input distributions have linear correlation. Spearman correlation assumes that the two input distributions have monotonic correlation. Kendall correlation assumes that the two input distributions have ordinal correlation (in ranks). The results suggest that the max distance among evidence has a negative correlation between the accuracy and the max distance among evidences, which can be interpreted as a lower accuracy is expected on the questions with more distantly distributed evidences.

Table 15: Two evaluator models' (GPT-4o-mini and Claude-3.5-sonnet) evaluations on the answers of these two models. The results show minimal variance between evaluators, with Claude-3.5 being slightly stricter but showing no significant model preference.

|  | Evaluated By GPT-4o-mini | Evaluated By Claude-3.5-sonnet |
|---|---|---|
| Claude-3.5-sonnet | 62.30 | 61.39 |
| GPT-4o-mini | 52.32 | 51.56 |

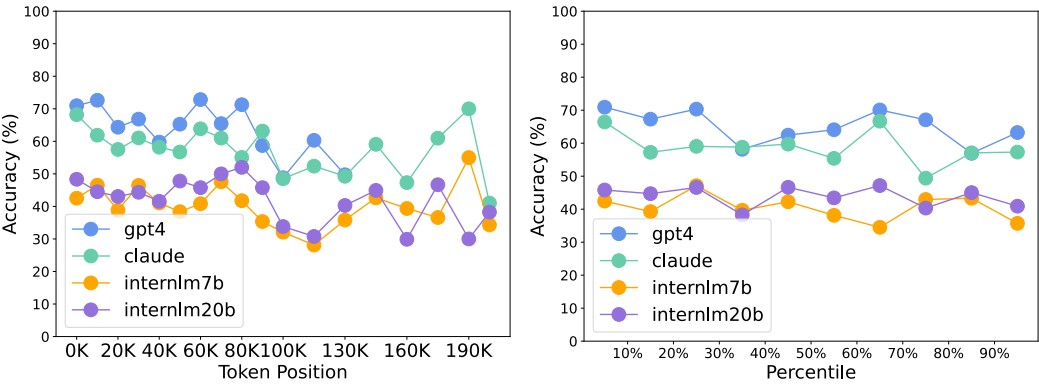

Figure 6: Analysis of Accuracy in multichoice Setting by Absolute and Relative Token Positions: This figure illustrates the accuracy in the multichoice setting of NovelQA, plotted against the absolute position (left) and percentage position (right) of each question's relevant evidence within the novel. Each subplot represents a different model. The x-axis of the absolute position figure (left), reflecting token indexes, is folded on the right due to the long-tail distribution in the lengths of the selected novels.

### C.4.6 REPRESENTATIVE ERRORS

Through reviewing the generation above, we concluded three common error types: hallucination, miscounting, and overlooking. Examples for each type are presented in Table 17.

**Hallucination** refers to the information generated by the model with factual errors. In our generative QA setting, typical hallucination mistakes encompass two types: (1) factual errors about the fictional settings (e.g., mixing entities within the setting or between the settings of different books) and (2) factual errors about the narrations (e.g., whether a fact is narrated). The first category usually appears in questions asking minor characters, plots, and settings, where the model might output a non-existing one. Meanwhile, the second category is often associated with sentence-locating questions, which ask the model to locate a sentence. In this case, the model may fake a sentence that does not exist in the original text.

**Overlooking** refers to the model's neglect of details. As mentioned above, the questions in *detailed* category involve minor characters, plots, or settings. Diving further, the reasons why these details are difficult to be recalled lie in two aspects: (1) They do not contribute to the character development, other plots, or the main themes, and thus reading the rest of the novel does not help to remind this detail; (2) Since most novels have derivative works (e.g., films, fan works, and book reviews), where the detailed information is eliminated to form a condensed narration. As the derivative works spread further and appear more frequently in the model's training data, they have a higher probability of becoming the models' inner knowledge, which is similar to (Chang et al., 2023)'s observation, and vice versa for those omitted details. These two factors contribute to the difficulty in the model's recalling details and thus result in overlooking errors.

**Miscounting** Researchers (Li et al., 2023a; Feng et al., 2023) have revealed shortcomings in the counting ability of LLMs, especially autoregressive-decoder-based models, and methods unfolding the outputs such as chain-of-thought prompting can enhance their counting ability. Our test does show that models make mistakes with numbers. Though the errors in the case of generative responses

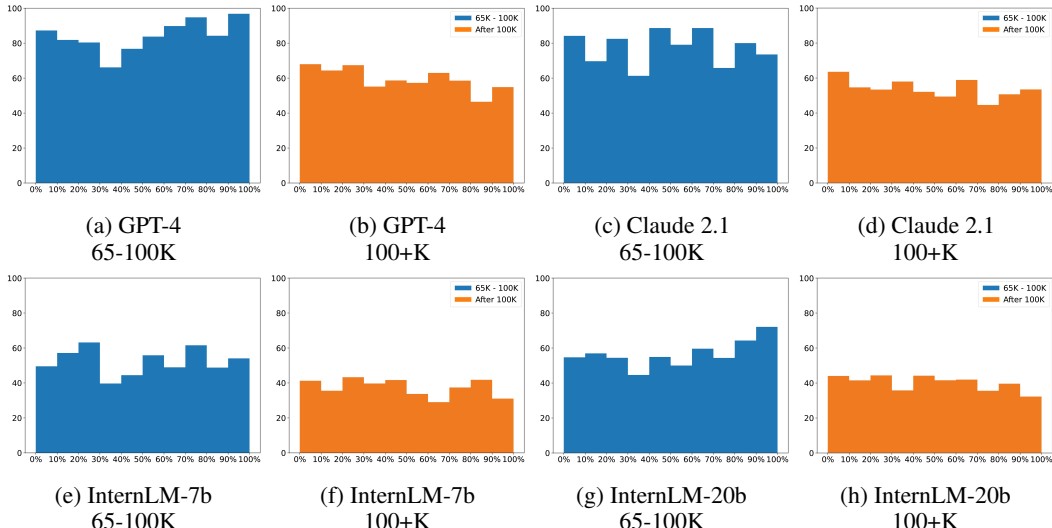

Figure 7: Performance Analysis by Relative Positions within Different Token Ranges: This figure illustrates the accuracy of various long-context LLMs in the multichoice setting, segmented by novels' length categories: 65K(the lowest token count) to 100K tokens and over 100K tokens. It highlights the models' performance trends relative to the evidence position within these ranges, showcasing the 'lost-in-middle' phenomenon in shorter texts and a performance decline towards the end in longer texts, thus revealing distinct comprehension challenges faced by LLMs when processing texts of varying lengths.

Table 16: Correlation between the distance among multiple evidences in multi-hop questions and the accuracy in the generative scenario. The max distance among all evidence distances are considered as the evidence distance of the question. The overall negative correlations show that the evidence distances are negatively correlated with the accuracies. The interpretation of matrices can be found in Appendix C.4.5.

| Model | Pearson | Spearson | Kendall |
|---|---|---|---|
| GPT-4 | -0.0734 (0.0879) | -0.2511 ($3.0696 \times e^{-09}$) | -0.2031 ($4.6129 \times e^{-09}$) |
| Claude-3 | -0.0120 (0.7899) | -0.1826 (4.6228) | -0.1385 ($9.0407 \times e^{-05}$) |
| InternLM-7b | -0.1140 (0.0083) | -0.2287 ($8.8721 \times e^{-08}$) | -0.1841 ($1.2663 \times e^{-07}$) |
| InternLM-20b | -0.0634 (0.1407) | -0.2017 ($2.1954 \times e^{-06}$) | -0.1626 ($2.722 \times e^{-06}$) |

may be due to not following instructions and simply outputting 'multiple times' instead of the desired specific times, the accuracy in the multichoice setting has still only reached 38.53% to 49.56% for the chosen four models, as shown in Table 17. Even in the simplest question, which asks for the appearing frequency of certain phrases, the models still make mistakes.

Table 17: Categories of representative errors observed in evaluating LLMs on NovelQA.

| Fault Type | Subtype | Setting | Example QA |
|---|---|---|---|
| **Hallucination** | On Fictional Setting | GPT-4, generative | **Book**: *Mansfield Park*
**Q**: What is the relationship between Miss Maria Ward and Lady Bertram?
**Correct A**: Miss Maria Ward and Lady Bertram are the same person.
**Model A**: Sisters. |
| | On Narration | GPT-4, generative | **Book**: *The Night Land*
**Q**: How many times has Aesworpth shouted?
**Correct A**: 1
**Model A**: Not mentioned. |
| **Overlooking** | | GPT-4, generative | **Book**: *Light in August*
**Q**: Who is Percy Grimm?
**Correct A**: Percy Grimm - the captain of the State National Guard who kills Joe Christmas and castrates him.
**Model A**: Percy Grimm does not appear in the novel. |
| **Miscounting** | | GPT-4, generative | **Book**: *Can You Forgive Her?*
**Q**: Has the word or phrase 'take away another man' appeared? If so, how many times does it appear in the text?
**Correct A**: Yes, 2.
**Model A**: Yes, 1. |

