# OpenReview forum: "NovelQA: Benchmarking Question Answering on Documents Exceeding 200K Tokens"
_ICLR.cc/2025/Conference — ICLR 2025 Poster_

### Official Review · Reviewer_RTuY · 2024-10-30

**Soundness:** 4
**Presentation:** 3
**Contribution:** 3
**Rating:** 8
**Confidence:** 4

**Summary:**

This paper introduces NovelQA, a benchmark designed to evaluate LLMs on long-context question-answering tasks. NovelQA addresses limitations in existing benchmarks by focusing on texts exceeding 200,000 tokens, derived from a diverse selection of English novels. NovelQA incorporates human-annotated questions that require detailed comprehension of lengthy narratives. The dataset evaluates multiple aspects of comprehension, including multi-hop reasoning, detail retrieval, and narrative coherence, revealing significant gaps in LLMs' ability to manage large, complex contexts.

**Strengths:**

NovelQA benchmark addresses an important challenge in long-context LLMs by introducing a dataset that emphasises extended context comprehension within complex narratives, representing a significant advancement over standard benchmarks. The dataset is carefully constructed with expert annotations, which are detailed in the paper, ensuring that question complexity aligns appropriately with model capabilities. The paper provides comprehensive documentation on question types and evaluation metrics, establishing a strong foundation for interpretability and contributing to a reliable benchmark. Last but not least, NovelQA evaluates narrative comprehension, multi-step reasoning, and retrieval-based tasks in LLMs, marking a substantial contribution to long-context evaluation.

**Weaknesses:**

Though the transparency of the annotation process is addressed, it could be strengthened. Although the authors state that questions are created by expert annotators with backgrounds in English Literature, the paper lacks clarity regarding annotator instructions, experience levels, and quality control measures above basic inter-annotator agreement scores — this could be demonstrated more clearly. Secondly, evaluation methods are insufficiently justified and may introduce bias due to the use of automated scoring. For instance, gpt-4 is used as an evaluator, with limited details on its evaluation process, which is particularly concerning given that the benchmark aims to capture complex reasoning across extended narratives.  eg. gpt-4 as an evaluator may score higher for gpt-4's answer. This could lead to bias in the results. While two human evaluators are mentioned, the paper does not provide information on their backgrounds or expertise. Other than these issues, the paper is well-written!

**Questions:**

1. I’m curious about the performance of other open-source models, such as mistral, llama, and so on, or perhaps smaller closed-source models like GPT-4o-mini. Could you provide additional insights into how these models perform compared to the primary models evaluated in the paper?

2. Regarding the use of gpt-4 as an evaluator, as noted above, this may introduce potential bias. Could you clarify how gpt-4 , or an alternative model, acted as an evaluator in the paper?

3. Are the questions and sample sentences in the dataset accessible online for user verification in the future, or are they restricted and not publicly available for sourcing and comparison?

---

> ### Author Response · Authors · 2024-11-21
> **Response to Reviewer RTuY**
>
> **Weakness1:** Though the transparency of the annotation process is addressed, it could be strengthened. Although the authors state that questions are created by expert annotators with backgrounds in English Literature, the paper lacks clarity regarding annotator instructions, experience levels, and quality control measures above basic inter-annotator agreement scores — this could be demonstrated more clearly.
>
> **Response to Weakness1:**
>
> Thanks for suggesting increased transparency in our annotation process documentation.
> We would like to highlight that these details are already included in our paper, though we acknowledge we can make them more prominent.
> - **Annotator Instruction**: Section 3.2, Annotator Recruitment. Annotators are allowed to select novels for annotation based on their familiarity, ensuring they had previously read and comprehensively understood the texts. Meanwhile, we make sure the selected books meet our standards of enough word count and well-developed narratives. We also ensure that each selected novel is either annotated by only one individual, or consistent in version across annotators, despite minor variations among different editions. Each annotator contributes to a typically small number of 20-30 questions per novel. This approach avoids forcing annotators annotating questions on unfamiliar content.
> - **Experience levels**: Section 3.2, Annotator Recruitment. Our annotators are predominantly English Language and Literature university students or those with a keen interest in English novels, recruited from local universities. These students are accustomed to closely reading English novels for coursework or as a hobby, and writing reports or papers on them.
> - **Quality control**: Section 3.2, Quality Control. Besides IAA, we involved at least five rounds of double-checking offered by the authors. The created data is manually double-checked by three authors of this work. The review procedure follows the criteria of minimizing factual errors, enhancing expression clarity, and maintaining challenges to LLMs. Consequently, only 79.4% of the collected data are preserved, resulting a final dataset of 2305 QA tuples.
>
> We will revise our paper to make these details more prominent and clearly structured.
>
> **Weakness2:**
> Secondly, evaluation methods are insufficiently justified and may introduce bias due to the use of automated scoring. For instance, gpt-4 is used as an evaluator, with limited details on its evaluation process, which is particularly concerning given that the benchmark aims to capture complex reasoning across extended narratives. eg. gpt-4 as an evaluator may score higher for gpt-4's answer. This could lead to bias in the results. While two human evaluators are mentioned, the paper does not provide information on their backgrounds or expertise.
>
> **Question2:** Regarding the use of gpt-4 as an evaluator, as noted above, this may introduce potential bias. Could you clarify how gpt-4 , or an alternative model, acted as an evaluator in the paper?
>
> **Response to Weakness2&Question2:**
>
> We appreciate the reviewer's important question about potential evaluation bias.
>
> We conducted a thorough analysis comparing different evaluator models to check their pontential:
>
> | |Evaluated By GPT-4o | Evaluated By Claude-3.5 |
> |------|------|------|
> |Claude-3.5-sonnet | 62.30 | 61.39 |
> |GPT-4o-mini | 52.32| 51.56 |
>
> The results show minimal variance between evaluators, with Claude-3.5 being slightly stricter but showing no significant model preference. This consistency is largely due to NovelQA's objective nature, with well-defined questions and answers that leave little room for evaluator bias.
>
> Regarding the background and expertise of two human evaluators:
>
> The two human evaluators are chosen from our top annotators, with an expertise level as Section 2.3 Section 3.2, Annotator Recruitment suggests. Our annotators are predominantly English Language and Literature university students or those with a keen interest in English novels, recruited from local universities. These students are accustomed to closely reading English novels for coursework or as a hobby, and writing reports or papers on them.These two evaluators were specifically chosen for their superior expertise and demonstrated thoroughness among all annotators.
>
>   We will enhance our paper's methodology section to more explicitly detail these evaluation procedures and evaluator qualifications, ensuring full transparency of our validation process.

---

> > ### Comment · Reviewer_RTuY · 2024-11-25
> >
> > Hi, I will be reevaluating the scores. Before doing so, could you clarify the differences between using GPT-4o as a judgment here and [1]?
> >
> > "Compared to humans, we do observe that some judges favor certain models. For example, GPT-4 favors itself with a 10% higher win rate; Claude-v1 favors itself with a 25% higher win rate. However, they also favor other models and GPT-3.5 does not favor itself. Due to limited data and small differences, our study cannot determine whether the models exhibit a self-enhancement bias. Conducting a controlled study is challenging because we cannot easily rephrase a response to fit the style of another model without changing the quality."
> >
> > [1] Zheng, L., Chiang, W.L., Sheng, Y., Zhuang, S., Wu, Z., Zhuang, Y., Lin, Z., Li, Z., Li, D., Xing, E. and Zhang, H., 2023. Judging llm-as-a-judge with mt-bench and chatbot arena. Advances in Neural Information Processing Systems, 36, pp.46595-46623.

---

> ### Author Response · Authors · 2024-11-21
> **Response to Reviewer RTuY (Part2)**
>
> **Question1:**
> I’m curious about the performance of other open-source models, such as mistral, llama, and so on, or perhaps smaller closed-source models like GPT-4o-mini. Could you provide additional insights into how these models perform compared to the primary models evaluated in the paper?
>
> **Response to Question1:**
> We sincerely thank the reviewer for this suggestion. We have expanded our evaluation to include the latest models with regards to all reviewers' comments :
>
> |  | Multi-choice | Generative |
> | ------------------|--------------------|----------------------- |
> | Claude-3.5-sonnet | 77.92 | 62.30
> | GPT-4o-mini | 71.85 | 52.32 |
> | Llama-3.1-8B |  62.31 | 42.65 |
> | Llama-3.1-70B |  69.39 | 51.50 |
>
> (We also attempted to evaluate Mistral-2-Large-2411, but it often failed to answer questions when processing prompts exceeding 100k tokens, both through local deployment and official API calls. So we believe  Mistral-2-Large-2411 may not be suitable for long-context question answering.)
>
> While these recent models show notable improvements compared to their predecessors, they still face significant challenges on NovelQA. We will include these updated results and their analysis in the revised paper.
>
>
> **Question3:** Are the questions and sample sentences in the dataset accessible online for user verification in the future, or are they restricted and not publicly available for sourcing and comparison?
>
> **Response to Question3:**
> We sincerely thank the reviewer for raising this important question about data accessibility.
>
> To maintain the integrity of NovelQA as a benchmark and prevent data leakage of golden answers and evidence, we have implemented a balanced approach to data access.
> 1. The questions can be accessed freely in Huggingface datasets.
> 2. We have established an automated evaluation platform (Codabench) that allows researchers to evaluate their models while protecting the golden answers from public exposure.
> 3. For researchers conducting in-depth analysis, we provide access to golden answers and evidence upon request, with the understanding that this data will be used for research purposes only and not be redistributed.
> 4. This access policy is clearly documented on our Codabench platform, GitHub repository and Huggingface dataset page.
>
> These approaches help prevent the training data from being inadvertently included in model training while ensuring the benchmark remains verifiable and useful for the research community.
>
> We will revise the paper to make these access policies more explicit.

---

> ### Author Response · Authors · 2024-11-25
> **Response to Reviewer RTuY**
>
> We sincerely thank the reviewer for this important question about potential evaluation bias compared to previous work.
>
> The key difference lies in the nature of questions being evaluated.
>
> The study by Zheng et al. (2023) focused on subjective and open-ended questions like "How does it affect my daily life? Give 3 examples." and "What are some business etiquette norms when doing business in Japan?". These questions have no standard answers, allow multiple valid perspectives, and often require longer, opinion-based responses. This subjectivity creates more room for model preferences to affect evaluation.
>
> In contrast, NovelQA primarily consists of objective questions, such as "How many times has Kitty kissed Walter? in The Painted Veil" and "In which year does the earliest event happen, and in which year does the latest event happen? in Tess of the d'Urbervilles." (more examples can be found in  Appendix Table 9). These questions have clear, verifiable answers with a factual basis in the text. This objectivity significantly reduces the impact of model bias in LLM-as-Judge evaluation, as the evaluator's task is more about verifying factual accuracy than judging subjective quality.
>
> We will elaborate on this distinction in our revised paper to provide a clearer explanation of why our evaluation methodology is robust against model bias.

---

> > ### Comment · Reviewer_RTuY · 2024-11-27
> > **Reviewer Response: Score Adjustment**
> >
> > I have raised the score for the paper. Thank you for your clarifications, and all the best!

---

### Official Review · Reviewer_Y5jR · 2024-10-31

**Soundness:** 3
**Presentation:** 3
**Contribution:** 2
**Rating:** 6
**Confidence:** 3

**Summary:**

This paper presents a new benchmark to assess LLMs on reading and understanding very long inputs (200k in average). Concretely, the task is Question Answering (QA) in two forms, multiple-choice and generative, over novels. The dataset is constructed based on public and copyrighted novels from the Project Gutenberg (purchasing ones when necessary) and manually crafted questions, gold answers (distractors generated automatically but manually inspected), and evidence. Human annotators are Language and Literature students. The benchmark contains questions of different complexity (multi-hop, single-hop, and details) and about different aspects (e.g., relationships or narrative settings -- 8 categories in total). Five large-context LLMs are evaluated on the benchmark (3 commercial and 2 open source). Results reveal that LLMs struggle to read, find and process the content in the long context necessary to formulate the answer; this happens even with commercial LLMs (GPT-4) with open-source LLMs exhibiting worse performance.

The paper is easy to follow and with illustrative examples. It includes performance analysis per question type and by content position.

**Strengths:**

The proposed benchmark is a valuable resource for the evaluation of LLMs performance on reading and reasoning to answer questions over long texts.

**Weaknesses:**

While the evaluation of LLMs shows that these struggle to understand, attend, and recall all the content from the long context necessary to answer questions (main focus of the paper), I wonder whether an initial extractive step would be a strong baseline in this setup. For instance, a common practice in summarisation of long inputs, is an initial extractive step (e.g., by tf-idf) that is applied before carrying out the actual task with the selected text. Maybe adding such a baseline would contribute to highlight the robustness of the proposed QA benchmark (i.e., how difficult is it to solve the task in this way). Also, some of the questions (e.g., counting) could be a LLM reasoning weakness rather than a long context processing issue.

**Questions:**

The authors could incorporate experiments with the new Llama 3.1 family.

---

> ### Author Response · Authors · 2024-11-21
> **Response to Reviewer Y5jR**
>
> **Weakness:** While the evaluation of LLMs shows that these struggle to understand, attend, and recall all the content from the long context necessary to answer questions (main focus of the paper), I wonder whether an initial extractive step would be a strong baseline in this setup. For instance, a common practice in summarisation of long inputs, is an initial extractive step (e.g., by tf-idf) that is applied before carrying out the actual task with the selected text. Maybe adding such a baseline would contribute to highlight the robustness of the proposed QA benchmark (i.e., how difficult is it to solve the task in this way). Also, some of the questions (e.g., counting) could be a LLM reasoning weakness rather than a long context processing issue.
>
> **Response to the Weakness:**
> We sincerely thank the reviewer for extractive baselines and the nature of question types. Let us address both points:
> Regarding the extractive baseline suggestion: We have actually conducted experiments with TF-IDF based extraction in both settings of NovelQA:
> 1. For the multiple-choice setting, a pure TF-IDF approach is not applicable as it cannot perform the reasoning required to select between options.
> 2. For the generative setting, we implemented a sophisticated TF-IDF baseline that:
> - Extracts relevant sentences using TF-IDF similarity
> - Identifies key phrases using multiple features:
>   - TF-IDF similarity scores
>   - Question keyword coverage
>   - Phrase length penalties
>
> The TF-IDF approach achieved only 0.3% accuracy.
>
> We also experimented with RAG (Retrieval-Augmented Generation) baselines before, which achieved 30-40% accuracy in generative setting, still significantly below long-context GPT-4 and Claude 2.1.
>
> Regarding counting questions: We intentionally included complex reasoning tasks rather than simple extraction tasks to evaluate long-context capabilities. This design choice helps assess models' ability to not just retrieve but also reason over extended contexts. The performance gap thus primarily reflects challenges in comprehensive long-context processing rather than just retrieval limitations.
>
> **Suggestion:** The authors could incorporate experiments with the new Llama 3.1 family.
>
> **Response to the Suggestion:**
> We sincerely thank the reviewer for this suggestion. We have expanded our evaluation to include the latest models with regards to all reviewers' comments :
>
> |  | Multi-choice | Generative |
> | ------------------|--------------------|----------------------- |
> | Claude-3.5-sonnet | 77.92 | 62.30
> | GPT-4o-mini | 71.85 | 52.32 |
> | Llama-3.1-8B |  62.31 | 42.65 |
> | Llama-3.1-70B |  69.39 | 51.50 |
>
> (We also attempted to evaluate Mistral-2-Large-2411, but it often failed to answer questions when processing prompts exceeding 100k tokens, both through local deployment and official API calls. So we believe  Mistral-2-Large-2411 may not be suitable for long-context question answering.)
>
> While these recent models show notable improvements compared to their predecessors, they still face significant challenges on NovelQA. We will include these updated results and their analysis in the revised paper.

---

> ### Comment · Reviewer_Y5jR · 2024-11-27
>
> Thank you for the clarification about the extractive -based variants and the question types. I have a follow-up question on the extractive + generative I meant in my comments (what you called RAG in the response?). With which LLM was the experiment(s) done? I think such a baseline should be included to highlight how difficult is to find the pieces of evidence (passages) needed to elaborate the answer.

---

> ### Author Response · Authors · 2024-11-27
> **Reply to Reviewer Y5jR**
>
> We appreciate the follow-up question about our RAG experiments.
>
> Here's how we implemented the RAG approach:
>
> We split each novel into chunks of 512-1024 tokens, ensuring natural breakpoints (e.g., line breaks or sentence endings) rather than arbitrary truncation. For each question, we use the BM25 algorithm to retrieve 10 relevant passages and include them in the prompt. We tested this with both GPT-4 and Claude-2.1 in the generative setting, achieving accuracies of 39.8% and 34.1% respectively. These results are significantly lower than their long-context counterparts (46.9% and 46.0%).
>
> This performance gap demonstrates that even with sophisticated retrieval methods, finding and reasoning over the correct passages remains challenging, further validating the robustness of NovelQA.
>
> We will include this analysis in our revised paper.

---

> ### Comment · Reviewer_Y5jR · 2024-11-27
>
> The reviewer thanks the authors for providing additional clarifications. The concerns about the extractive baseline and the difficulty of questions/finding the information were addressed, thus the reviewer raised the score accordingly. It would be useful to include the discussion about the RAG experiments in the final version of the paper.

---

### Official Review · Reviewer_EUJu · 2024-11-01

**Soundness:** 3
**Presentation:** 3
**Contribution:** 3
**Rating:** 6
**Confidence:** 4

**Summary:**

NovelQA is a new benchmark designed for evaluating large language models (LLMs) on complex, extended narratives with average context windows exceeding 200,000 tokens. This is relevant as the most advanced long-context LLMs can process over 250,000 tokens. NovelQA provides a combination of complexity, length, and narrative coherence, using a diverse selection of English novels from various eras, genres, and formats. Professional expert annotators carry out the annotation process, and all of them hold or are pursuing degrees in English Literature. The dataset includes multi-hop, single-hop, and detail questions, which assess the model's abilities to retrieve and integrate scattered information, summarize information, and accurately identify specific and subtle details. Both closed APIs and open models have been evaluated using this benchmark, and a comprehensive analysis is proposed based on the results.

**Strengths:**

— Annotators performed a huge amount of work. The questions, golden answers, and evidence of NovelQA are crafted through the efforts of experts. The context is extensive, making it challenging to create such a set.

— An informative, descriptive Appendix with details, annotation agreements, and error analysis.

— The contribution is clear and high

**Weaknesses:**

— No ethical consideration and Limitations sections. The Limitation section is highly recommended, as mentioning the restrictions is important.

— There are efficiency problems with the benchmark running for the long-context models.

— To add in the limitations: data leakage. `To prevent against data leakage, we will not release golden answers for the test set, minimizing the risk of overfitting.`
The novels are not created from scratch. Many of them, particularly public domain works, are already included in the training datasets of models like GPT-4. This represents an indirect issue of data leakage. It is also crucial to discuss the problems related to data contamination and leakage.

— The biases of the annotators also need to be mentioned in the Limitation section.

— Truncation is also a limitation as nobody checked whether the truncated part influences the result.

— The creation of templates based on the most difficult cases for GPT-4 and Claude2.1 is understandable but may cause biases.

— Syntactic choices for the multiple choices:
`We use GPT-4 to generate three distracting options for each question and its golden answer and randomly permute the four answers`
Authors further evaluate API models on the same sets: it will be with a high probability of a strong bias for GPT4 to answer the questions.
Thus, the authors can not claim that the GPT performs better than others.

**Questions:**

— Somehow the Table 7 and Figure 5 are above the Appendix section, and needs to be formatted.

— No Table number in Line 1120

— Consider using VLLM models to reduce running costs and time.

— Better to write `multichoice` or `multi-choice` consistently during the paper

— Add Human Baseline in Table 2

— Write in bold the best scores in Tables, better readability

---

> ### Author Response · Authors · 2024-11-21
> **Response to Reviewer EUJu (Part1)**
>
> **Weakness1**: No ethical consideration and Limitations sections. The Limitation section is highly recommended, as mentioning the restrictions is important.
>
> **Response to Weakness1**:
> We sincerely apologize for this oversight. Due to an inadvertent error during submission, our Limitations and Ethics Statements sections were accidentally commented out in this version, although they are present in our public version. We will restore these important sections in the revised paper. Here are the original sections:
>
> \section{Limitations}
>
> Access to Close-source LLMs: One significant limitation is our inability to obtain APIs for certain close-source long-context LLMs, such as Baichuan-192K, GLM4-200K, and Moonshot-192K.
>
> Technical Constraints for Open-source LLMs: The majority of open-source LLMs are not optimized for processing inputs exceeding 128K tokens with high GPU memory cost. This technical challenge renders direct inference on NovelQA infeasible for these models. Consequently, our experiments were limited to LLMs that have been adapted using LMDeploy.
> [Note: While this limitation was initially present, recent experiments using vLLM have largely addressed this concern. We will update this section accordingly in the revision.]
>
> Language Limitation: NovelQA, and all associated data are exclusively in English.
>
> [New Added] genre Limitation: NovelQA focus on the novel genre.
>
> \section{Ethics Statements}
> 1. We are dedicated to ensuring that NovelQA serves exclusively for academic and scientific endeavors. We have  launched an evaluation website, a leaderboard website, and the provision of an API for data access. As certain novels used in our project are protected by copyright, we affirm that we will not release these novels.
> 2. NovelQA does not contain personally identifiable information or offensive content.
>
> We have featured these sections in the revised paper and addressed potential concerns and constraints of our work. Thank you for bringing this to our attention - it helps maintain the rigor and transparency of our research.
>
> **Weakness2**: There are efficiency problems with the benchmark running for the long-context models.
> **Suggestion3**: Consider using VLLM models to reduce running costs and time.
>
> **Response to Weakness2 and Suggestion3**: We appreciate the suggestion about vLLM. In our recent experiments, we have successfully implemented vLLM, which has significantly reduced both GPU memory usage and inference time. This optimization allows us to run our benchmark more efficiently. We will update the paper to reflect these improvements in computational efficiency.
>
> **Weakness3**:
> To add in the limitations: data leakage. To prevent against data leakage, we will not release golden answers for the test set, minimizing the risk of overfitting. The novels are not created from scratch. Many of them, particularly public domain works, are already included in the training datasets of models like GPT-4. This represents an indirect issue of data leakage. It is also crucial to discuss the problems related to data contamination and leakage.
>
> **Response to Weakness3:**
> We appreciate the reviewer's important point about data leakage.
> We should clarify that our discussion of data leakage specifically refers to the golden answers of NovelQA. While we acknowledge that some novels in our dataset (especially public domain works) may have been included in LLMs' training data, our primary concern is preventing NovelQA's golden answers from appearing in LLMs' pretraining or supervised fine-tuning datasets. This is crucial because:
> 1. Novel content exposure doesn't guarantee understanding of specific question-answer patterns
> 2. Our test set's golden answers remain private to prevent direct optimization
> 3. The complexity of our questions requires genuine reasoning rather than pattern matching
>
> We will revise our paper to make this distinction clearer and discuss both aspects of data exposure in both main text and Limitations section.
>
>
> **Weakness4:**  The biases of the annotators also need to be mentioned in the Limitation section.
>
> **Response to Weakness4:** We appreciate the reviewer's concern about potential annotator biases. In our annotation process, annotators were specifically instructed to focus on objective questions only. We also implemented multiple quality control steps to identify and remove subjective questions (See section 3.2 Quality Control). We will soon include our annotation guideline regarding ensuring objectiveness and minimizing the bias in Section 3 - Data Collection and Annotation to clarify this question. We will also enhance the Limitations section to detail these measures and acknowledge any remaining potential for annotator bias.

---

> ### Author Response · Authors · 2024-11-21
> **Response to Reviewer EUJu (Part2)**
>
> **Weakness5:** Truncation is also a limitation as nobody checked whether the truncated part influences the result.
>
> **Response to Weakness5:** Thank you for addressing the critical point regarding truncation.
> Truncation is a standard practice for adapting longer texts to fit within the constrained context windows of LLMs, especially when evaluating their performance on long-context tasks. While it's possible that some evidence may fall within the truncated parts, we view the limited context window as a current limitation of the models, rather than a flaw in our dataset. Thus, we did not specifically avoid placing evidence within these sections. We are optimistic that future LLMs, exemplified by Claude and Gemini, which boast a 1M context window size, will overcome these limitations and fully encompass all contextual evidence.
>
> **Weakness6:** The creation of templates based on the most difficult cases for GPT-4 and Claude2.1 is understandable but may cause biases.
>
> **Response to Weakness6:**
> Thanks for raising this  concern about potential biases in our template design process.
> While we acknowledge this methodological consideration, we would like to clarify our approach and its rationale:
> - Human control in question design: Although the error patterns are inspired by GPT-4 and Claude2.1, we manually picked those that are meaningful to be evaluated on (see Table 10 for question type - purpose). Human control exists in this procedure.
> - Error cases are general: The fact that the designed evaluation set shows even less accuracy (see Table 2) on other models (InternLM-7b and internLM-20b) demonstrates the generalizability of the observed errors.
> We will address the issues above in a limitation section, ensuring transparency about potential biases and their mitigation strategies
>
> **Weakness7:** Syntactic choices for the multiple choices: We use GPT-4 to generate three distracting options for each question and its golden answer and randomly permute the four answers Authors further evaluate API models on the same sets: it will be with a high probability of a strong bias for GPT4 to answer the questions. Thus, the authors can not claim that the GPT performs better than others.
>
> **Respose to Weakness7:** We appreciate the reviewer's concern about potential bias in multiple-choice questions. Our pilot study compared distractors generated by both GPT-4 and Claude 2.1. Interestingly, GPT-4 generated slightly more challenging distractors - both models scored approximately 0.5% lower on GPT-4's distractors compared to Claude 2.1's. This indicates no advantageous bias for GPT-4. We will add this pilot study analysis to the paper to address this methodological concern.
>
> **Remaining Suggestions**:
> — Somehow the Table 7 and Figure 5 are above the Appendix section, and needs to be formatted.
> — No Table number in Line 1120
> — Better to write multichoice or multi-choice consistently during the paper
> — Add Human Baseline in Table 2
> — Write in bold the best scores in Tables, better readability
>
> **Response to Remaining Suggestions**: We sincerely thank Reviewer EUJu for the detailed and in-depth suggestions. We will fix these problems, thanks!

---

### Official Review · Reviewer_E41W · 2024-11-04

**Soundness:** 3
**Presentation:** 3
**Contribution:** 3
**Rating:** 6
**Confidence:** 3

**Summary:**

This paper presents a new dataset on long-context understanding. It uses novel as the long context and evaluate the model with questions from different complexity and aspect categories. Those questions and answers are annotated by human annotators, with a high inter-annotator agreement on answers. The answers are also with distractions for multi-choice setting generated from GPT-4. They evaluate open-source and commercial LLMs on the constructed dataset, revealing some findings such as performance degradation after 100K tokens, difficulty with questions on meaning, relation, span and times and low performance on evidences recall.

**Strengths:**

1. This paper presents a new dataset on long-context understanding. It has some unique features that are not covered by previous work, such as the long averaged context length, and human annotation efforts. It has the potential to be used in later work.

2. The benchmarking results include some interesting findings, such as performance degradation after 100K tokens and low performance on evidence recall. Those findings may enlighten future research in this area,

3. This paper is well-organized and easy to follow, with comprehensive appendix on the details of the data.

**Weaknesses:**

1. One major concern of this work is that this paper only focuses on using novels as the genre for long-context understanding. It would be better if the author could cover other long-context genres, such as other nonfiction books, to evaluate the long-context understanding comprehensively.

2. Another concern I have is about the analysis of the performance based on the position of corresponding evidence. Because the annotation on the annotation does not include whether the evidence is unique in the long context, there may be some cases where the annotated evidence can also be found in another snippet of the context. And it may ultimately impact the analysis or the conclusion of the analysis based on the evidence position.

3. As we can see in Table 1, most of the questions are either single-hop or details, which is generally based on a short text snippet of the long context. It would be also beneficial to include more discussion on the multi-hop questions, for example, the performance with regard to the relative distances of the pieces of evidence for a multi-hop question, as these questions are more difficult to be produced by LLMs.

**Questions:**

1. How are the question types and evidence annotated?

2. What does the distribution of the evidence positions look like?

---

> ### Author Response · Authors · 2024-11-21
> **Response to Reviewer E41W (Part1)**
>
> **Weakness1**: One major concern of this work is that this paper only focuses on using novels as the genre for long-context understanding. It would be better if the author could cover other long-context genres, such as other nonfiction books, to evaluate the long-context understanding comprehensively.
>
> **Response to Weakness1**:
>
> We appreciate the reviewer's suggestion about including diverse text genres. We would like to clarify our rationale for focusing on novels in this initial benchmark:
> 1. Coherence and Complexity: As mentioned in Lines 051-052, novels were specifically chosen because "they are long and complex, with plots that are closely linked from start to end." This inherent narrative continuity provides a rigorous test of models' ability to maintain coherent understanding across long contexts.
> 2. Annotation Quality: The annotation of novels requires relatively less domain expertise compared to specialized texts like legal or financial documents, enabling us to maintain high annotation quality while keeping the benchmark creation process practical and scalable.
> 3. Natural Temporal and Causal Dependencies: Novels naturally contain rich interdependencies between characters, events, and plot developments, making them ideal for evaluating models' ability to track and reason about long-range relationships in text.
> We fully acknowledge the importance of evaluating long-context understanding across different genres. This novel-based benchmark serves as our first phase, and we are actively working on extending our framework to include other forms of long-context text in future work, such as technical documents and specialized professional texts.
>
> We will include this discussion in the Limitations section.
>
> **Weakness2:** Another concern I have is about the analysis of the performance based on the position of corresponding evidence. Because the annotation on the annotation does not include whether the evidence is unique in the long context, there may be some cases where the annotated evidence can also be found in another snippet of the context. And it may ultimately impact the analysis or the conclusion of the analysis based on the evidence position.
>
> **Response to Weakness2:**
>
> We  appreciate the reviewer's thoughtful concern about evidence position analysis and potential duplicate evidence in long contexts.
> We have carefully designed our annotation process to address this concern:
> As Table 1 suggests, the multi-hop questions are spanning over variance Aspect labels.
>   - On span, setting, relation and character classes, evidences are ensured to be sufficient -- otherwise, the conclusion of 'over which years is the story spanning ' 'in which places happens the story' 'the relationship among characters' will result in factual errors, which are easy to be found and correct during double-checking.
>   - On times, meaning, and plot classes, we tried to encourage the annotators to annotate the objects or events that appear very little, so that the evidences are countable. Given this less workload, we encourage the annotators to retrieve as sufficient evidences as they can.

---

> > ### Comment · Reviewer_E41W · 2024-11-26
> >
> > **Weaknesses 1**
> >
> > Thanks for the explanations. The point is not the rationale behind choosing novels as the corpus for long-context understanding, but whether novels are representative enough for a comprehensive evaluation. In that sense, I still think it is kind of limited.
> >
> > **Weaknesses 2**
> >
> > I think there is probably a misunderstanding here. What I would like to talk about is whether the occurrence times may impact the analysis. For example, Figure 4 shows that the model tends to find the answer more effectively when the token position is early in the sequence. But could it be that when the answer appears early in the context, it is more likely to appear multiple times afterward, particularly more so than when an answer first occurs later in the context? If that is true, then what is the deciding point, the occurrence times, or the position?

---

> ### Author Response · Authors · 2024-11-21
> **Response to Reviewer E41W (Part2)**
>
> **Weakness3:** As we can see in Table 1, most of the questions are either single-hop or details, which is generally based on a short text snippet of the long context. It would be also beneficial to include more discussion on the multi-hop questions, for example, the performance with regard to the relative distances of the pieces of evidence for a multi-hop question, as these questions are more difficult to be produced by LLMs.
>
> **Response to Weakness3**:
>
> In fact, multi-hop questions constitute 35.0% (807/2305) of the dataset - a substantial portion, as shown in Table 1.
> Besides,  single-hop and detail questions also often require comprehensive context understanding rather than just short snippet extraction:
> 1. They demand deep comprehension of specific plot points or character development
> 2. Models must precisely locate relevant information within 200k tokens
> 3. The challenge lies in both understanding and retrieval across long contexts
>
> Regarding  *the performance with regard to the relative distances of the pieces of evidence for a multi-hop question*, we also conducted an experiment to test the relationship of token distance and accuracy. We define the evidence distance (in number of tokens since the book starts) of a multi-hop problem as the max distance among all its related evidences. Then we calculated the correlation between the answer's correctness ((0,1)-distribution) on the generative task and the evidence distance on each model.
>
> | Model | Pearson |Spearman |Kendall|
> |---------|---------|---------|----------|
> |GPT-4 |-0.0734 (0.0879)|-0.2511 (3.0696e-09)|-0.2031 (4.6129e-09)|
> |Claude 3 |-0.0120 (0.7899) |-0.1826 (4.6228)|-0.1385 (9.0407e-05)|
> |InternLM-7b |-0.1140 (0.0083) |-0.2287 (8.8721e-08) | -0.1841 (1.2663e-07)|
> |InternLM-20b |-0.0634 (0.1407) |-0.2017 (2.1954e-06) |-0.1626 (2.722e-06)|
>
> Among the indices,
> - Pearson correlation: assumes that the two input distributions have linear correlation.
> - Spearman correlation: assumes that the two input distributions have monotonic correlation.
> - Kendall correlation: assumes that the two input distributions have ordinal correlation (in ranks)
> The results suggest that the max distance among evidence has a negative correlation between the accuracy and the max distance among evidences, which can be interpreted as a lower accuracy is expected on the questions with more distantly distributed evidences.
>
> We will add this analysis to the revised paper to provide deeper insights into question difficulty and model performance.

---

> > ### Comment · Reviewer_E41W · 2024-11-26
> >
> > **Weakness3**
> >
> > Thanks for the analysis on multihop settings. It is clear and intuitive.

---

> ### Author Response · Authors · 2024-11-21
> **Response to Reviewer E41W (Part3)**
>
> **Question1**: How are the question types and evidence annotated?
>
> **Response to Question1**:
>
> We sincerely thank the reviewer for this important question about our annotation process.
> Question Type Annotation:
> - Initial classification: Based on question templates and key words/phrases
> - Verification: Manual double-checking to ensure accuracy
>
> This two-step process helps maintain consistent classification while catching potential edge cases
>
> Evidence Annotation:
> - Primary Annotation Method: Annotators were instructed to directly copy relevant sentences from the original text that support their answers. This ensures that evidence is firmly grounded in the source material and maintains traceability
> - Evidence Format: 1. Standard format: Direct quotes from the original text; 2.Extended format (in specific cases): "<original sentence>, in Chapter <n>, <brief explanation of reasoning>"
> - Quality Control: Annotators could request permission to include brief explanatory notes when direct quotes alone might not fully convey their reasoning. These requests were individually reviewed to maintain annotation consistency while ensuring comprehensiveness
>
> We will provide more detailed documentation of these processes in our revised paper.
>
> **Question2:** What does the distribution of the evidence positions look like?
>
> **Response to Question2:**
>
> Thanks for raising the concern about the distribution of the evidence positions.
>
> Of the 3,666 evidence instances, 134 are summaries or comments without exact matches in the document. The distribution of the remaining 3,532 instances is as follows:
> Absolute distribution (by absolute token index):
>
> |Range| 0~20K | 20～40K | 40～60K | 60～80K | 80～100K | 100～130K | 130～170K | 170～210K | >210K |
> | --| -- | -- | -- | -- | -- | -- | -- | -- | -- |
> |Count| 959 | 631 | 446 | 359 | 249 | 239 | 214 | 209 | 226 |
> |Percentage| 27.15 | 17.87 | 12.63 | 10.16 | 7.05 | 6.77 | 6.06 | 5.91 | 6.40 |
>
> Approximately 18.4% of the evidence is located beyond the 130K token mark (outside GPT-4's context window), and 6.4% is beyond 210K tokens (outside Claude and InternLM's context windows).
>
> Given the varying lengths of novels, we also analyzed the relative position distribution (by relative token position: token position / total tokens in the corresponding book):
>
> | Range | 0~10 | 20 | 30 | 40 | 50 | 60 | 70 | 80 | 90 | 100 |
> | -- | -- | -- | -- | -- | -- | -- | -- | -- | -- |  -- |
> | Count |  889 | 431 | 360 | 326 | 288 | 271 | 274 | 250 | 205 | 238 |
> | Percentage | 25.17 | 12.20 | 10.19 | 9.23 | 8.15 | 7.67 | 7.76 | 7.08 | 5.80 | 6.74 |
>
> Our analysis reveals that the highest proportion of evidence instances (25.17%) occurs within the first 10% of the novels, while the distribution across the middle sections is relatively uniform. This concentration in the early parts of the novels can be attributed to the initial introduction of characters and plot elements. The first occurrence of evidence related to these introductions naturally falls in the earlier sections of the novels. It's important to note that during the question formulation process, we did not deliberately adjust the distribution of questions. The observed pattern in answer locations emerges naturally from the narrative structure of the novels.
>
> We will incorporate these analyses into our revision to provide a more comprehensive understanding of the dataset's characteristics and potential implications for model evaluation.

---

> ### Author Response · Authors · 2024-11-28
> **Reply to Reviewer E41W**
>
> **Response to Follow-up Weakness 1**:
>
> We sincerely appreciate the reviewer's perspective. We agree that novels alone cannot provide a complete evaluation of long-context understanding. While NovelQA serves as a foundational work that reveals fundamental challenges in long-context processing (such as information retrieval, temporal reasoning, and relationship tracking), we recognize the need for broader genre coverage. We are actively extending our framework to other domains, including academic papers, technical documentation, and legal documents. That said, we believe the insights gained from NovelQA have broader implications, as many of the core challenges in long-context understanding - such as maintaining coherence across long distances and resolving complex references - are common across different text genres.
>
> **Response to Follow-up Weakness 2**:
>
> We appreciate the reviewer's thoughtful hypothesis about the potential correlation between evidence position and repetition frequency. However, our empirical results suggest this is likely not a major confounding factor:
>
> As shown in Figure 4 and Table 4, model performance remains relatively stable across the 0-100K token range, only showing significant degradation after the 100K position. If repetition frequency were a dominant factor, we would expect to see notably higher performance in the first 20K tokens, where initial introductions of recurring information typically appear in novels.
>
> While we acknowledge that evidence repetition might play a role in some cases, the observed performance pattern suggests that position effects are more possible to be driven by models' intrinsic limitations in processing very long contexts.

---

### Official Review · Reviewer_aWGC · 2024-11-05

**Soundness:** 3
**Presentation:** 3
**Contribution:** 3
**Rating:** 6
**Confidence:** 4

**Summary:**

This paper introduces a new benchmark called NovelQA, which is designed to evaluate the performance of LLMs on extremely long and complex texts. NovelQA uses English novels as contexts and annotates questions across three levels of complexity and seven aspects by literature experts. The proposed benchmark surpasses existing ones in length, includes evidence alongside questions, and emphasizes detailed comprehension. This paper also presents experimental results from a suite of LLMs on this benchmark and analyses their performance across different aspects.

**Strengths:**

1. The paper is generally well-written and clear.
2. The proposed NovelQA benchmark features the longest context to date
3. The experiments are generally well-designed

**Weaknesses:**

1. The benchmark only focuses on novels, which is somewhat limited. It is essential to include other forms of long-context in order to provide a more comprehensive evaluation of LLMs' long-context understanding capabilities.
2. The templates mainly focus on information extraction types of tasks. It would be beneficial to design more complex questions, e.g. ones
that require reasoning

**Questions:**

1. How was the evidence annotated?
2. It was mentioned in the paper that LLMs "usually fail to tackle information spanning over multiple chapters." How does the proposed benchmark assess this aspect?

---

> ### Author Response · Authors · 2024-11-21
> **Responses to Reviewer aWGC**
>
> **Weaknesses1**: The benchmark only focuses on novels, which is somewhat limited. It is essential to include other forms of long-context in order to provide a more comprehensive evaluation of LLMs' long-context understanding capabilities.
>
> **Response to Weaknesses1**:
>
> We appreciate the reviewer's suggestion about including diverse text genres. We would like to clarify our rationale for focusing on novels in this initial benchmark:
>
> 1. Coherence and Complexity: As mentioned in Lines 051-052, novels were specifically chosen because "they are long and complex, with plots that are closely linked from start to end." This inherent narrative continuity provides a rigorous test of models' ability to maintain coherent understanding across long contexts.
>
> 2. Annotation Quality: The annotation of novels requires relatively less domain expertise compared to specialized texts like legal or financial documents, enabling us to maintain high annotation quality while keeping the benchmark creation process practical and scalable.
>
> 3. Natural Temporal and Causal Dependencies: Novels naturally contain rich interdependencies between characters, events, and plot developments, making them ideal for evaluating models' ability to track and reason about long-range relationships in text.
> We fully acknowledge the importance of evaluating long-context understanding across different genres. This novel-based benchmark serves as our first phase, and we are actively working on extending our framework to include other forms of long-context text in future work, such as technical documents and specialized professional texts. We will include this discussion in the Limitations section.
>
> **Weakness2**: The templates mainly focus on information extraction types of tasks. It would be beneficial to design more complex questions, e.g. ones that require reasoning
>
> **Response to Weakness2**:
>
> In fact, the vast majority of NovelQA questions require complex reasoning rather than simple extraction. Let us elaborate with specific examples:
>
> 1. Complex Reasoning Requirements:
> We take three types of questions as examples that require complex reasoning:
> - Times Questions: Require models to retrieve multiple facts, integrate them, and perform reasoning
> - Relation Questions: Demand reasoning about character relationships and their evolution throughout the narrative, going beyond simple fact retrieval
> - Span Questions: Necessitate complex temporal reasoning to identify and compare multiple timestamps
>
> 2. Comprehensive Ability Assessment: As detailed in Appendix Table 10, our question types evaluate a wide range of cognitive abilities:
>
> | Question Type | Abilities Required |
> |----------------|--------------|
> | multi-hop | Ability to retrieve and integrate scattered information.|
> | single-hop | Ability of information retrieval. |
> | detailed | Abilities to precisely identify specific, subtle details. |
> | times | Abilities to retrieve facts, integrate facts, and reason. |
> | meaning | Ability to retrieve ambiguous information and reason. |
> | span, setting, relation | Abilities to retrieve facts, integrate facts, and reason. |
> | Character, plot | Abilities to subtle or ambiguous information and reason. |
>
> 3. Statistical Evidence:
> - Only ~4% of questions can be answered through simple text search
> - These questions serve as "Needle in the Haystack" evaluations, a standard benchmark practice
> - The remaining 96% require various forms of complex reasoning
>
> 4. Our questions are specifically designed to evaluate both:
> - Complex reasoning capabilities of LLMs
> - Practical applicability in real-world scenarios
>
> We will revise our paper to make these details more prominent and clearly structured.

---

> ### Author Response · Authors · 2024-11-21
> **Response to Reviewer aWGC (part2)**
>
> **Question1**: How was the evidence annotated?
>
> **Response to Question1**:
>
> We sincerely thank the reviewer for this important question about our annotation process.
> We followed a rigorous manual annotation process for evidence collection. Here are the specific details:
> Primary Annotation Method:
> - Annotators were instructed to directly copy relevant sentences from the original text that support their answers
> - This ensures that evidence is firmly grounded in the source material and maintains traceability
>
> Evidence Format:
> - Standard format: Direct quotes from the original text
> - Extended format (in specific cases): "<original sentence>, in Chapter <n>, <brief explanation of reasoning>"
>
> Quality Control:
> - Annotators could request permission to include brief explanatory notes when direct quotes alone might not fully convey their reasoning
> - These requests were individually reviewed to maintain annotation consistency while ensuring comprehensiveness
> We will add detailed documentation of this annotation process in the revised paper for complete transparency
>
> **Question2:**  It was mentioned in the paper that LLMs "usually fail to tackle information spanning over multiple chapters." How does the proposed benchmark assess this aspect?
>
> **Response to Question2:**Since most multi-hop questions span over multiple chapters, the annotators are instructed by the annotation guideline to find multi-hop questions with evidences spanning more distantly, which is further ensured by authors' checking.
> For example, in the demonstration book Frankenstein ,
> - How many times has Robert written letters to his sister?  The answer ‘11 times’∂ is supported by evidence across 1k ~ 90k tokens, which must be spanning across chapters.
> - In which chapter(s) is/are Safie mentioned? Safie is mentioned multiple times across chapters.
> Thus, we consider that our multi-hop question evaluates the information-retrieval across chapters.
>
> The relationship can be roughly estimated through their low performance on multi-hop questions (See Table 11).
> Besides, we also added an experiment to test the relationship of token distance and accuracy. We define the evidence distance (in number of tokens since the book starts) of a multi-hop problem as the max distance among all its related evidences. Then we calculated the correlation between the answer's correctness ((0,1)-distribution) on the generative task and the evidence distance on each model.
>
> | Model | Pearson |Spearman |Kendall|
> |---------|---------|---------|----------|
> |GPT-4 |-0.0734 (0.0879)|-0.2511 (3.0696e-09)|-0.2031 (4.6129e-09)|
> |Claude 3 |-0.0120 (0.7899) |-0.1826 (4.6228)|-0.1385 (9.0407e-05)|
> |InternLM-7b |-0.1140 (0.0083) |-0.2287 (8.8721e-08) | -0.1841 (1.2663e-07)|
> |InternLM-20b |-0.0634 (0.1407) |-0.2017 (2.1954e-06) |-0.1626 (2.722e-06)|
>
> Among the indices,
> - Pearson correlation: assumes that the two input distributions have linear correlation.
> - Spearman correlation: assumes that the two input distributions have monotonic correlation.
> - Kendall correlation: assumes that the two input distributions have ordinal correlation (in ranks)
> The results suggest that the max distance among evidence has a negative correlation between the accuracy and the max distance among evidences, which can be interpreted as a lower accuracy is expected on the questions with more distantly distributed evidences.
>
> We will include the analysis in our revised paper.

---

### Author Response · Authors · 2024-11-25
**New Revision and General Response**

Dear Reviewers,

We sincerely thank you for your thoughtful comments and constructive suggestions that have helped strengthen our paper.

We have submitted our responses and the revised manuscript with changes highlighted in red.

As the discussion phase ends in two days, we kindly request confirmation of receipt and welcome any additional feedback on our responses.

Thank you for your time and consideration.

Sincerely,

The Authors

---

### Meta-Review · Area_Chair_uG3X · 2024-12-23

**Metareview:**

The paper presents NovelQA, a benchmark of question answering where the context can be of length up to than 200k tokens.  The benchmark is quite thorough, the paper well written and the reviewers are all positive wrt the paper.  The paper also benchmarks many LLMs on this task which is a valuable set of datapoints.  Given this, I suggest acceptance.

**Additional Comments On Reviewer Discussion:**

There has been a significant discussion between authors and reviewers for this paper that improved both mutual understanding of the paper and a better assessment for it.

---

### Decision · Program_Chairs · 2025-01-22

Accept (Poster)